# Small deviations in kinematics and body form dictate muscle performances in the finely tuned avian downstroke

Marc E Deetjen[1], Diana D Chin[1], Ashley M Heers[1,2], Bret W Tobalske[3], David Lentink[1,4]*

[1]Department of Mechanical Engineering, Stanford University, Palo Alto, United States; [2]Department of Biological Sciences, California State University, Los Angeles, United States; [3]Division of Biological Sciences, University of Montana, Missoula, United States; [4]Faculty of Science and Engineering, University of Groningen, Groningen, Netherlands

**Abstract** Avian takeoff requires peak pectoralis muscle power to generate sufficient aerodynamic force during the downstroke. Subsequently, the much smaller supracoracoideus recovers the wing during the upstroke. How the pectoralis work loop is tuned to power flight is unclear. We integrate wingbeat-resolved muscle, kinematic, and aerodynamic recordings *in vivo* with a new mathematical model to disentangle how the pectoralis muscle overcomes wing inertia and generates aerodynamic force during takeoff in doves. Doves reduce the angle of attack of their wing mid-downstroke to efficiently generate aerodynamic force, resulting in an aerodynamic power dip, that allows transferring excess pectoralis power into tensioning the supracoracoideus tendon to assist the upstroke—improving the pectoralis work loop efficiency simultaneously. Integrating extant bird data, our model shows how the pectoralis of birds with faster wingtip speed need to generate proportionally more power. Finally, birds with disproportionally larger wing inertia need to activate the pectoralis earlier to tune their downstroke.

*For correspondence:
d.lentink@rug.nl

## eLife assessment

This **important** study combines experiments and mathematical modeling to enhance our understanding of the interplay between the two flight muscles in birds during slow flight. The evidence for the findings is **compelling**, derived from new methods for measuring wing shape and force production combined with previously validated methods in muscle physiology. This work will be of broad interest to comparative biomechanists.

## Introduction

Understanding how birds use their flight muscles to power flight is key to understanding their biomechanics and movement ecology, as well as skeletal muscle performance in general. During slow flight, negligible lift is induced by forward body motion, so instead, almost all the aerodynamic force is generated by the flapping motion of the wings. Hence, slow flight is one of the most energetically expensive modes of locomotion (*Altshuler and Dudley, 2003*; *Ellerby and Askew, 2007*; *Tobalske et al., 2003*; *Heers et al., 2016*). Furthermore, in generalist birds, the downstroke muscle, the pectoralis, is of primary importance energetically: it is the largest muscle in the avian body. It is 3–5 times larger than the next largest wing muscle, the supracoracoideus, which controls the upstroke (*Tobalske and Biewener, 2008*). Considering the pectoralis dominates power production during the stroke of

the avian forelimb—in contrast to the distributed muscle groups that typically move vertebrate limbs (*Liem et al., 2001*; *De Luliis and Pulera, 2019*)—the avian pectoralis serves as an ideal model system for understanding peak skeletal muscle performance. Finally, mastering the first few downstrokes during takeoff represents critical steps toward full flight in both fledgling birds and presumably avian precursors, so better understanding the factors which dictate how the downstroke of an adult bird functions can inform our understanding of the development of avian flight.

Because of the multifaceted importance of pectoralis function, much prior research has been conducted to measure the time-resolved power produced by this muscle *in vivo*. However, technological limitations have left our understanding incomplete (*Ellerby and Askew, 2007*; *Tobalske et al., 2003*; *Tobalske and Biewener, 2008*; *Biewener et al., 1992*; *Biewener et al., 1998*; *Jackson and Dial, 2011a*; *Dial et al., 1997*; *Soman et al., 2005*; *Ingersoll and Lentink, 2018*). Measurements of both muscle stress and strain are needed to directly measure pectoralis power. However, the pectoralis has a broad origin with a complex arrangement of muscle fibers, making global measurement of muscle fiber strain challenging. The current state-of-the-art recordings in birds measure muscle length changes with sonomicrometry (*Biewener et al., 1998*; *Soman et al., 2005*). To measure muscle stress, the state-of-the-art solution is to attach a strain gauge to the deltopectoral crest (DPC), which is located where the pectoralis inserts on the humerus (*Tobalske et al., 2003*; *Tobalske and Biewener, 2008*; *Biewener et al., 1992*; *Biewener et al., 1998*; *Dial and Biewener, 1993*). However, pull calibrations to correlate bone strain to muscle stress are complicated by multiple factors. One source of skew is misalignment between the changing pull axis of the muscle through the stroke *in vivo* and the pull axis of the force transducer used for the pull calibration *post mortem* (*Tobalske and Biewener, 2008*), which was refined by *Jackson and Dial, 2011a*. Another calibration method for correlating muscle power with aerodynamic and inertial power integrated over a wingbeat cycle relies on highly simplified aerodynamic models (*Hedrick et al., 2003*). An *in vitro* alternative relies on electrical stimulation of bundles of muscle fibers (*Ellerby and Askew, 2007*; *Askew and Marsh, 2001*), but the heterogeneous composition of the pectoralis leads to variability (*Reiser et al., 2013*), and *in vivo* validation is lacking. Hence, fundamental questions about *in vivo* muscle function remain (*Biewener, 2011*).

In contrast to the muscle architecture of the pectoralis, which is well suited for producing work to flap the wing, the function of its antagonist muscle, the supracoracoideus, is more ambiguous. It has pennate muscle fibers attached to a long tendon, making it better suited for producing force than work, which would facilitate storing and releasing significant amounts of elastic energy in the supracoracoideus tendon (*Biewener and Baudinette, 1995*; *Roberts et al., 1997*; *Biewener, 1998*; *Biewener and Roberts, 2000*). Measurements in pigeons (*Columba livia*) suggest this elastic storage may range between 28 and 60% of the net work of the supracoracoideus (*Tobalske and Biewener, 2008*), which is thought to primarily elevate and supinate the wing. However, while the supracoracoideus is used in every flight mode (*Dial, 1992*; *Tobalske, 1995*), birds are still able to fly without its use (*Degernes and Feduccia, 2001*; *Sokoloff et al., 2001*). In summary, how the avian flight apparatus moves the wing by tuning pectoralis and supracoracoideus muscle and tendon power release—to generate the net aerodynamic force required for flight—has yet to be resolved.

To resolve how the avian pectoralis work loop is tuned to generate aerodynamic force, we combine established *in vivo* muscle strain and activation measurement techniques with our new *in vivo* high-speed 3D shape reconstruction and aerodynamic force measurement techniques (*Figure 1*), which we integrate with a new mathematical model derived from first principles. We used our high-speed, structured-light technique (*Deetjen et al., 2017*; *Deetjen and Lentink, 2018*) to 3D-reconstruct wing morphology and directly measured horizontal and vertical aerodynamic forces using our aerodynamic force platform (AFP) (*Chin and Lentink, 2017*; *Hightower et al., 2017*; *Lentink et al., 2015*; *Chin and Lentink, 2019*; *Deetjen et al., 2020*). We simultaneously measured the time-resolved pectoralis activation and strain *in vivo* using electromyography and sonomicrometry (*Tobalske et al., 2003*). We focused on the second downstroke after takeoff during level flight in ringneck doves (*Streptopelia risoria*; hereafter 'doves') because it was the first downstroke that well supported bodyweight after takeoff. First, we computed the time-resolved power and torque that the flight muscles need to sustain to overcome wing inertia and generate aerodynamic force. To account for elastic storage and understand how variations in the magnitude and timing of elastic storage in the supracoracoideus tendon would affect power generation in the pectoralis and supracoracoideus muscles, we derived a



**Figure 1.** Wingbeat-resolved aerodynamic forces, pectoralis activation and contraction, and 3D surface reconstruction of four doves (N = 4) in slow hovering flight (n = 5). (**A**) Inset showing the average second wingbeat (N × n = 20 flights total) net horizontal (x, purple) and vertical (z, orange) aerodynamic force, electrical activation (EMG, pink) of the left pectoralis (right pectoralis signal was unreliable), and strain of the left (pink) and right (blue) pectoralis. Gray region: downstroke; color shaded regions: standard deviation; force normalized by bodyweight: bw; statistics and plot definitions apply to all figures unless stated differently. At the top is shown a dove's 3D reconstructed surface and recorded aerodynamic force during each stroke phase (flight direction mirrored to match temporal direction). (**B**) The same data from (**A**) are plotted for a single representative flight from takeoff to landing. The dotted lines represent horizontal and vertical perch forces during takeoff and landing. In addition to the lowpass filtered (pink) EMG signal, the raw signal is plotted in black. (**C**) The total 3D aerodynamic force ($\vec{F}_{wing}$, black) is the sum of the measured horizontal (x) and vertical (z) components ($\vec{F}_{wing,x,z}$, gray) combined with the computed lateral (y) component (dotted line connecting $\vec{F}_{wing}$ and $\vec{F}_{wing,x,z}$). Using the 3D surface model (depicted at 17% of the second wingbeat), we illustrate the reconstruction of drag ($\vec{D}$, red) and lift ($\vec{L}$, blue) based on drag pointing opposite to wing velocity ($\vec{v}_{aero}$, black) and being perpendicular to lift. Drag and lift act perpendicular to the vector connecting the shoulder joint to the ninth primary wingtip ($\mathbf{X}_{P9}$). Along this vector, the mass of the wing is discretized using 20 point masses (green spheres; volume proportional to mass). Twelve body landmarks were manually tracked; black dots: ninth primary wingtip ($\mathbf{X}_{P9}$), seventh secondary feather, shoulder joint, wrist; gray dots: middle of the back (next to shoulder), left and right feet, left and right eyes, top of the head; gray cone: tip and base of the beak.

The online version of this article includes the following figure supplement(s) for figure 1:

**Figure supplement 1.** Experimental setup. Simultaneous measurement of the electrical activation and strain of the pectoralis muscle, the horizontal and vertical aerodynamic forces, and the 3D wing, body, and tail surface of doves in slow forward flight enabled us to reconstruct the aerodynamic work loop of the pectoralis.

**Figure supplement 2.** Comparison of aerodynamic and perch forces prior to surgery and after surgery shows that the surgery substantially altered the flight performance.

**Figure supplement 3.** Measured data is plotted for each of the four doves individually, averaged over five flights per dove.

muscle mechanics model from first principles to integrate our *in vivo* data. Finally, we used this model to gain comparative insight into muscle function by examining how the biomechanics of the downstroke scales across species and can explain avian flight muscle functionality generally.

## Results

The results are ordered in five sections that we integrate in the discussion to show how small deviations in kinematics and body form dictate muscle performances in the finely tuned avian downstroke. The 'Effects of angle of attack on aerodynamic power' section starts with the effect of angle of attack on aerodynamic power, and the 'Inertial versus aerodynamic power' section delineates how inertial versus aerodynamic power build up the total power output. In the 'Effects of elastic storage in the supracoracoideus tendon' section, we use muscle mechanics models to determine how the interplay between the pectoralis and supracoracoideus muscles is shaped by elastic storage in the supracoracoideus tendon and requires fine-tuning. We complete our mechanical analysis of the flight apparatus in the 'Directionality of pectoralis pulling on the humerus' section with the directionality of how the pectoralis pulls on the humerus to beat the wing. Finally, we generalize our flight muscle mechanics findings through a scaling analysis across extant birds in the 'Scaling analysis across extant birds' section and find that fine-tuning of the downstroke depends on scale.

### Effects of angle of attack on aerodynamic power

During the second wingbeat, there is a dip in the angle of attack during the middle of the downstroke when peak aerodynamic force is generated. This dip coincides with a peak in the power factor (non-dimensional lift$^{3/2}$ to drag ratio), which hence lessens the aerodynamic power required to generate lift. During the downstroke, the angle of attack across the span of the wing (*Figure 2C*, *Figure 2—figure supplement 1D*) starts and ends with peaks with a valley in-between: the first peak (54.0° ± 5.0°) is reached after 7.1% of the stroke and the second peak (68.4° ± 12.1°) is reached at 49.4% of the stroke (which corresponds to 10.9% before the end of the downstroke), with a large dip (30.5° ± 3.1°) in the middle at 27.7% of the stroke. Our aerodynamic force recordings show this dip in mid-downstroke angle of attack corresponds with a mid-downstroke dip in wing drag and peak in net aerodynamic force (*Figure 3B*). Comparing the angle of attack (*Figure 2C*) with the power factor (*Figure 3C*) of the doves during the downstroke, the power factor peak (timing after start of downstroke: 29.8% of the stroke) corresponds to the mid-downstroke angle of attack dip. Of the two peaks in drag (*Figure 3B*), the second peak after the mid-downstroke dip is larger (maximum drag scaled by bodyweight before mid-downstroke dip: 0.92 ± 0.22; after: 1.60 ± 0.46). This is caused by two factors: the corresponding angle of attack is higher and the wing area is larger. Examining the directional components of lift and drag (*Figure 3—figure supplement 1*), we find the dove primarily uses lift for weight support (stroke-averaged vertical aerodynamic force scaled by bodyweight for lift: 66.17% ± 10.81%; drag: 22.11% ± 4.62%). The secondary function of lift is to overcome drag (stroke-averaged horizontal aerodynamic force scaled by bodyweight for lift: 32.84% ± 9.96%; drag: –27.64% ± 4.64%). Drag primarily points backward throughout the downstroke, and laterally at the beginning and end of the downstroke (the lateral forces of the wings cancel). Due to the small peaks in drag before and after the mid-downstroke dip, drag contributes to weight support at the beginning and end of the downstroke, so both lift and drag contribute to weight support during takeoff. Finally, during the upstroke the dove folds its wings inward (*Figure 2—figure supplement 1A–C*), producing very little aerodynamic force (scaled by bodyweight vertically: 11.28% ± 2.81%; horizontally: 2.68% ± 1.49%).

Due to the mid-downstroke dip in drag, there is a corresponding dip in aerodynamic power (*Figure 4A*) because only the drag and wing velocity contribute to aerodynamic power by definition (*Equation S17*). The dip in aerodynamic power is less pronounced than the dip in drag because it corresponds with a peak in wing speed, but the uneven adjacent peaks remain (peak in aerodynamic power scaled by pectoralis mass before mid-downstroke dip: 513.0 ± 176.1 W/kg; after: 719.1 ± 175.0 W/kg; minimum dip: 460.4 ± 176.1 W/kg). This mid-downstroke dip in total aerodynamic power also corresponds to a dip in the lateral component of power, so that during mid-downstroke, the primary contribution to aerodynamic power is associated with the horizontal component of drag (*Figure 3—figure supplement 1A*) due to the forward movement of the wing (*Figure 2D*). As expected, the largest contribution to aerodynamic force is in the vertical direction (*Figure 3A*), yet when it comes

**Figure 2.** Measured wingbeat kinematics show wing area, extension, and speed are maximal mid-downstroke, while the wing angle of attack reaches a local minimum. (**A**) The stroke plane (light orange) fits the path of the vector originating from the shoulder joint and extending to the ninth primary feather tip ($\mathbf{X}_{P9}$) during the wingbeat, which we use to define wing kinematics with three angles: (1) the stroke plane angle $\phi_{stroke}$, (2) the wing stroke angle $\theta_{stroke}$, and (3) the deviation of the stroke angle from the stroke plane $\theta_{deviation}$. (**B**) The average stroke plane differs across the four doves while being consistent for each individual dove. (**C**) The wing stroke angle (orange), stroke deviation angle (green), and wing angle of attack (red) are consistent (low variance) across all flights and all four individuals. The only exception is the angle of attack during the upstroke. The avatar shows the angle of attack (red dotted arc), the angle between the wing chord (thin) and velocity vector (thick), at 17% of the second wingbeat (star in panels **C** and **D**). (**D**) The net wing speed (black) peaks mid-downstroke and mid-upstroke. The x-velocity component dominates net speed. The avatar shows the 3D wing velocity vector.

The online version of this article includes the following figure supplement(s) for figure 2:

**Figure supplement 1.** Stroke-resolved wing morphing parameters during the second wingbeat, measured using 3D wing surface reconstruction.

to aerodynamic power the vertical contribution is the smallest. This is because lift is much larger than drag in the vertical direction (*Figure 3—figure supplement 1C*), and only overcoming drag requires aerodynamic power (per the definition of lift and drag with respect to wing velocity).

Overall, we find that peak aerodynamic force production occurs mid-downstroke when drag attains a local minimum, yielding a peak in aerodynamic efficiency (highest power factor).

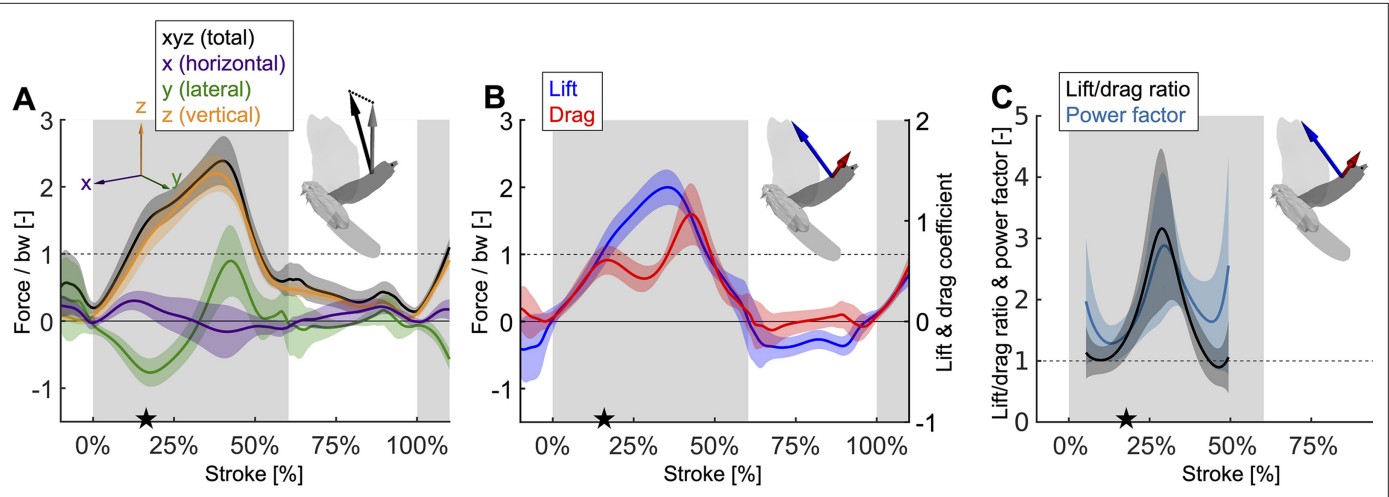

**Figure 3.** Mid-downstroke the lift peaks while drag reaches a local minimum, which coincides with a dip in the angle of attack (***Figure 2C***), causing the power factor (non-dimensional lift$^{3/2}$/drag) to reach a local maximum while the wing attains peak speed (***Figure 2D***). This reduces the aerodynamic power needed to generate peak lift during mid-downstroke and support bodyweight (bw; used to normalize force). (**A**) Net aerodynamic force magnitude (black) peaks mid-downstroke and is composed of the recorded large vertical (z) and small horizontal (x) force as well as the computed lateral (y) force. The avatar shows both the 2D measured aerodynamic force vector ($\vec{F}_{wing,x,z}$; gray) and the net 3D vector ($\vec{F}_{wing}$; black). (**B**) During mid-downstroke, the lift force (blue) peaks while the drag force (red) dips. Lift is defined positive with respect to the wing surface normal, hence negative lift opposes gravity during the upstroke because the wing is inverted. (**C**) The lift to drag ratio and power factor (non-dimensional lift$^{3/2}$/drag) peak mid-downstroke when force and velocity peak. Both ratios are not well-defined during stroke reversal and upstroke because both force and velocity are small, so these portions of the plot have been removed.

The online version of this article includes the following figure supplement(s) for figure 3:

**Figure supplement 1.** The components of lift and drag expand our understanding from ***Figure 3B***.

## Inertial versus aerodynamic power

Aerodynamic power dominates during the downstroke, whereas inertial power dominates the upstroke (***Figure 4B***; ***Supplementary file 1e***). However, we observe inertial power evens out the two peaks in aerodynamic power adjacent to the mid-downstroke dip (peak in aerodynamic + inertial power scaled by pectoralis mass before mid-downstroke dip: 537.0 ± 221.5 W/kg; after: 580.0 ± 176.8 W/kg; minimum dip: 455.0 ± 287.0 W/kg). Whereas for the aerodynamic power, the difference in the peaks is 206.1 ± 248.3 W/kg, with the addition of inertial power the difference in peaks reduces to 43.0 ± 283.4 W/kg.

## Effects of elastic storage in the supracoracoideus tendon

We use our time-resolved muscle model to understand the effect which the amount and timing of elastic storage in the supracoracoideus tendon has on power generation in the pectoralis and supracoracoideus muscles (***Figure 5***).

Varying elastic storage fraction in the supracoracoideus tendon reveals the power needed to tension the supracoracoideus tendon is small in comparison to the power generated by the pectoralis during the downstroke (***Figure 5***, ***Figure 5—figure supplement 1***, ***Supplementary file 1e***). The pectoralis would need to generate 206.0 ± 49.6 W/kg (stroke-averaged positive power scaled by pectoralis mass) if no energy was stored in the supracoracoideus tendon. At the other extreme, an increase of 24.9% ± 17.5% power generation in the pectoralis (to 255.9 ± 62.7 W/kg) would stretch the tendon of the supracoracoideus and allow it to store enough elastic energy to fully power the upstroke. Storing only a fraction mid-downstroke muscle work is still sufficient to partially power the upstroke (***Figure 5—figure supplement 1Q and R***). While our measurements and analysis cannot be used to deduce the energy storage fraction directly, anatomy (***Tobalske and Biewener, 2008***; ***Biewener and Baudinette, 1995***; ***Roberts et al., 1997***; ***Biewener, 1998***; ***Biewener and Roberts, 2000***) and a direct measurement of the strain in the tendon (***Tobalske and Biewener, 2008***) indicate a non-zero elastic energy storage fraction. Applying a storage fraction of 28–60% as measured in ***Tobalske and Biewener, 2008*** to our model requires only an increase in pectoralis work of between

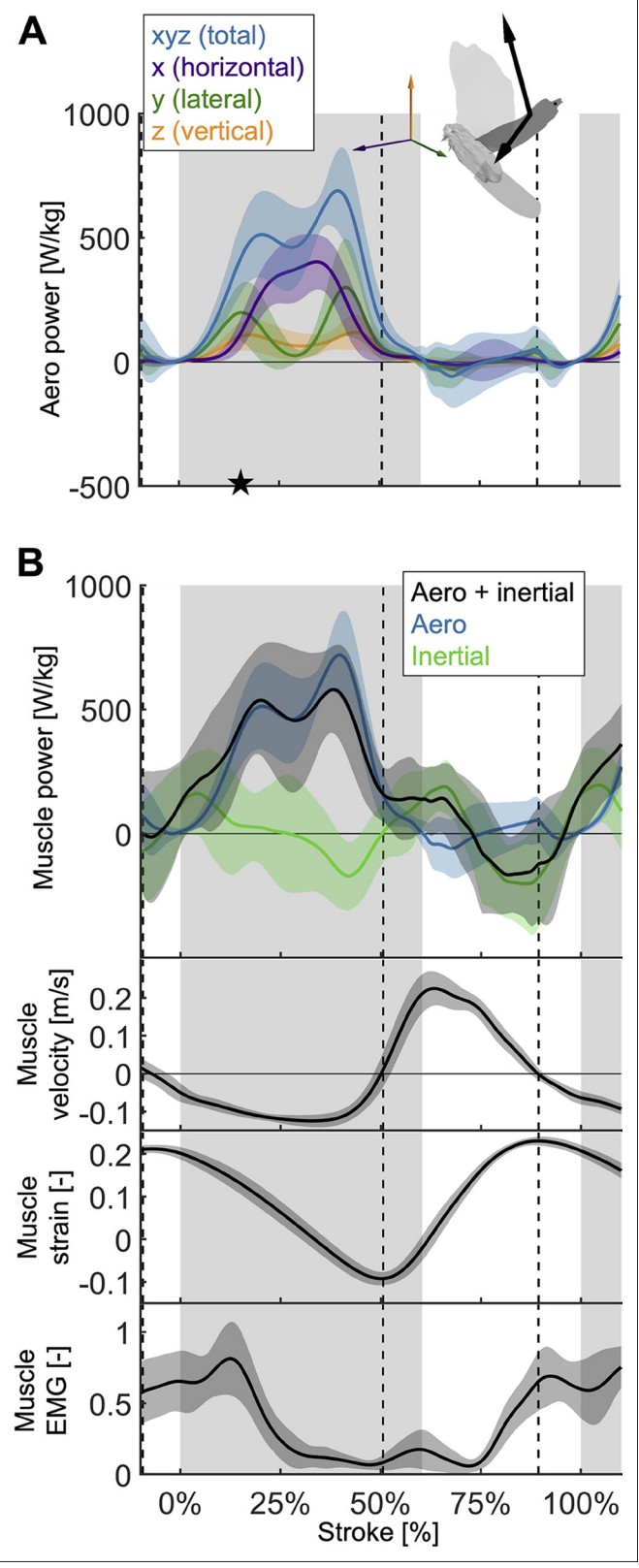

**Figure 4.** The total power (aerodynamic plus inertial) that the flight muscles need to generate dips midstroke. (**A**) The primary source of aerodynamic power stems from the horizontal contribution of drag (x, purple) because the wing moves forward (*Figure 2D*) while generating a dominant opposing drag force (*Figure 3—figure supplement 1A*). Because lift dominates drag in the vertical direction (*Figure 3—figure supplement 1C*), and

*Figure 4 continued on next page*

*Figure 4 continued*

lift acts perpendicular to wing velocity (making the dot product zero), the vertical component of aerodynamic power (z, orange) is the smallest despite large vertical aerodynamic force (**Figure 3A**). The avatar illustrates the wing velocity and net aerodynamic force vectors. (**B**) The total power (black) the flight muscles need to generate is dominated by the aerodynamic power (blue line) during the downstroke, and by the inertial power (green line) during the upstroke. The onset of electrical activation of the pectoralis (muscle EMG) lines up with the onset of pectoralis shortening (decreasing muscle strain, negative muscle velocity), which starts mid-upstroke. Notably, pectoralis shortening velocity reaches zero (dashed vertical line; also in subsequent plots) right before the start of the upstroke, when the required muscle power is low (and plateaus), and the supracoracoideus is known to take over (**Tobalske and Biewener, 2008**).

The online version of this article includes the following figure supplement(s) for figure 4:

**Figure supplement 1.** The three components of inertial power are plotted.

---

6.8 and 14.5%. Hence, elastic storage in the supracoracoideus tendon appears to be an effective solution to help power the upstroke that simultaneously smoothes the mid-stroke power output of the pectoralis.

Modeling variation in the timing of energy storage in the supracoracoideus tendon, we find that fine-tuning the timing improves the shape of the pectoralis work loop (**Figure 5**, **Figure 5—figure supplement 2**). Short storage time (**Figure 5C**) results in a spike in pectoralis power near the end of the downstroke, which produces a low pectoralis work loop shape factor (the actual loop area divided by the potential maximum area for the observed peak force and length change: 0.49 ± 0.14; **Figure 5E and G**). On the other hand, when the storage time is appropriately spread out over the downstroke (**Figure 5D**), the pectoralis work loop shape factor increases to a maximum value of 0.73 ± 0.11. While this maximum shape factor is achieved when the energy storage in the supracoracoideus tendon equals 90%, a shape factor of over 0.72 ± 0.11 can be achieved for any energy storage over 35% (**Figure 5F and G**). For any energy storage over 35%, the maximum shape factor corresponds to a storage time of 31% of the stroke. The reason for this is related to the mid-downstroke dip in required muscle power (**Figure 4B**) originating from the mid-downstroke dip in the angle of attack and drag (**Figure 2C**). When the energy storage in the supracoracoideus tendon is spread out during the downstroke, the peak in extra pectoralis power needed for tensioning overlaps with the dip in required muscle power from aerodynamic and inertial power. Hence, storing energy in the supracoracoideus tendon for 31% of the stroke has the effect of flatting out the mid-downstroke pectoralis power generation and improving the shape factor. Since a mid-contraction dip in power generation is unnatural for muscles, tensioning the supracoracoideus tendon is key for proper wing stroke kinematics: instead of the wing continuing to accelerate mid-downstroke when velocity peaks, the wing reaches zero acceleration at this point, after which it decelerates to prepare for stroke reversal.

## Directionality of pectoralis pulling on the humerus

Thus far, we have focused on power, which is a scalar, but in order to determine in what direction the pectoralis needs to pull on the humerus to effectively flap the wing, we need to expand our analysis to moments (torques) in three dimensions (**Figure 6**).

As is the case for the power analysis, aerodynamic moments dominate the net torque on the humerus during the downstroke, whereas inertia dominates during the upstroke and near stroke reversal (**Figure 6—figure supplement 1**). Notably, the combined vertical components of lift and drag (relative to gravity) dominate the torque because both are substantial (Section A3).

By combining the required muscle moment with the position of the humerus time-resolved, we can analyze the direction that the pectoralis pulls on the humerus during the downstroke (**Figure 6D, E**). For this, we assume that the muscle moment is primarily generated by the pectoralis, although this is likely not entirely true if the supracoracoideus tendon is tensioned during the mid-to-late portion of the downstroke. Under this assumption, for the entire downstroke, in the mediolateral direction, the pectoralis pulled medially (**Figure 6D**), and at the beginning of the downstroke, the pectoralis pulled in the cranioventral direction. At the end of the downstroke, the pull direction is in the caudodorsal direction, which indicates a contribution from the supracoracoideus.

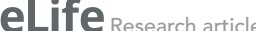

**Figure 5.** Finely tuned energy storage in the supracoracoideus tendon simplifies the function of both major flight muscles. (**A**) Our power model was derived by combining the external and internal force balance shown with the associated kinematic velocities (not shown); see equations in *Figure 7— figure supplement 1*. Approximately all internal flight forces and associated power are produced by the two primary flight muscles, the pectoralis (pink; $\vec{F}_{pect}$) and the supracoracoideus (orange; $\vec{F}_{supra}$), which move the mass of the body (green; $m_B$: modeled as a point mass) and wings (green; $m_W$: modeled as 20 point masses displayed in *Figure 1C*) to generate the wings' aerodynamic forces (blue; $\vec{F}_{aero}$) and thus power to sustain flight. (**B–D**) The pectoralis (pink) generates positive power during the downstroke and acts like a brake to absorb negative power during the late-upstroke (power is normalized by pectoralis mass). The supracoracoideus (orange) generates and/or releases positive power during the upstroke. Any energy released by the supracoracoideus tendon during the upstroke must first be stored during the downstroke, which is plotted as negative power. (**B**) Without energy storage in the supracoracoideus tendon, there is an unfavorable mid-downstroke dip in the required pectoralis output power. (**C**) If energy storage

*Figure 5 continued on next page*

*Figure 5 continued*

is poorly timed during the downstroke, the required pectoralis output power unfavorably spikes. (**D**) If the timing and quantity of energy storage are finely tuned, the pectoralis work loop is more favorably shaped mid-downstroke as it more closely approximates the theoretically ideal rectangular work loop (yellow rectangle in [**E**]) (**E**) The finely tuned elastic storage shown in (**D**) corresponds with the black work loop and results in the highest pectoralis shape factor (0.73) for the range of elastic storage parameters we simulated in (**F**), with stars corresponding to the loops in (**E**). Numbers between parenthesis in [**E**]: shape factor; dark gray shading: electrical pectoralis activation; light gray shading: downstroke phase. The shape factors of both the no-storage (brown: [**B**]) and finely tuned storage (black: [**D**]) cases exceed prior experimental shape factor measurements, of doves flying similarly slow (***Tobalske et al., 2003***). (**G**) Work loop shape factor as a function of elastic energy storage timing and fraction (color codes energy storage fraction): see *Figure 5—figure supplements 1–3* for a detailed overview.

The online version of this article includes the following figure supplement(s) for figure 5:

**Figure supplement 1.** Using our energy storage model (***Figure 5A***), we analyze how differing amounts of energy storage in the supracoracoideus tendon would affect how the total time-resolved required muscle power (***Figure 4B***) is split between the pectoralis and supracoracoideus.

**Figure supplement 2.** Using our energy storage model (***Figure 5A***), we find that spreading out the energy storage in the supracoracoideus tendon during the downstroke substantially improves the shape factor of the pectoralis muscle work loop by better flattening the mid-downstroke dip in required muscle power (***Figure 4B***), resulting in a peak shape factor of 0.73.

**Figure supplement 3.** Using our energy storage model (***Figure 5A***), we find that for supracoracoideus energy storage fractions above 35%, a pectoralis work loop shape factor of over 0.72 can be achieved by spreading the storage time in the supracoracoideus tendon out over 31% of the stroke.

## Scaling analysis across extant birds

Based on scaling the gross parameters related to aerodynamic power, we modeled a scaling relationship to predict average pectoralis power across extant birds (***Figure 7A and C***). We predict that average pectoralis power, $^{\mathrm{N}}\bar{P}^{\mathrm{pect}}$, divided by bodyweight, $m_{\mathrm{body}}$, scales linearly with flapping frequency, $f$, and wingspan, $|\vec{\mathbf{r}}_{\mathrm{span}}|$ (from the shoulder joint to the wingtip):

$$\frac{^{\mathrm{N}}\bar{P}^{\mathrm{pect}}}{m_{\mathrm{body}}} = -0.59 + 18.74 \left( f|\vec{\mathbf{r}}_{\mathrm{span}}| \right) \tag{1}$$

To compare the effects of aerodynamics and inertia on muscle power during the downstroke of level flight, we modeled a separate scaling relationship to predict the required timing of pectoralis power across extant birds (***Figure 7B and D***). Although inertial power does not contribute to average muscle power across the downstroke, because the positive and negative inertial power cancels, it does have the effect of shifting the distribution of required muscle power earlier in the stroke. Hence, for birds where the ratio of inertial to aerodynamic power is higher, the pectoralis muscle needs to provide power earlier in the downstroke. Quantitatively we predict that the midway point of the pectoralis power exertion, $T_{\mathrm{P,mid}}^{\mathrm{pect}}$ (***Equation S75***; percentage of stroke), equals

$$T_{\mathrm{P,mid}}^{\mathrm{pect}} = 35.01 - 61.75 \left\{ \frac{m_{\mathrm{wing}}}{m_{\mathrm{body}}} \left( \frac{r_{\mathrm{gyr}}}{|\vec{\mathbf{r}}_{\mathrm{span}}|} \right)^2 \left( f|\vec{\mathbf{r}}_{\mathrm{span}}| \right) f \right\}, \tag{2}$$

where $m_{\mathrm{wing}}$ is the mass of the wing, and $r_{\mathrm{gyr}}$ is the radius of gyration of the wing. A similar relationship holds for the midway point of the pectoralis force exertion, $T_{\mathrm{F,mid}}^{\mathrm{pect}}$ (***Equation S76***; percentage of stroke):

$$T_{\mathrm{F,mid}}^{\mathrm{pect}} = 31.90 - 59.06 \left\{ \frac{m_{\mathrm{wing}}}{m_{\mathrm{body}}} \left( \frac{r_{\mathrm{gyr}}}{|\vec{\mathbf{r}}_{\mathrm{span}}|} \right)^2 \left( f|\vec{\mathbf{r}}_{\mathrm{span}}| \right) f \right\} \tag{3}$$

Across a wide range of bird sizes (between 9.5 g and 2140 g), the variation in predicted pectoralis timing is small. Pectoralis midway power timing, $T_{\mathrm{P,mid}}^{\mathrm{pect}}$, falls between 23.0 and 32.7% and force timing, $T_{\mathrm{F,mid}}^{\mathrm{pect}}$, falls between 20.6 and 29.6%.



**Figure 6.** To quantify how the recruitment of the pectoralis muscle changes during the downstroke, we consider the balance of 3D angular momentum by calculating the associated torques. (**A**) The forces are as in our 1D power balance (**Figure 5A**), but instead of considering the entire bird, we evaluate the balance around the left shoulder joint. Consequently, body mass becomes irrelevant, and instead a reaction force ($\vec{F}_0$; cancels out) and moment ($\vec{M}_0$; assumed small compared to other torques) at the shoulder joint appear. (**B**) The pull angle between the humerus and the pectoralis, $\theta_p$, dictates how effectively the pectoralis exerts torque on the wing. The inset shows the corresponding lateral view with subscript 'b' referencing the body frame while no subscript indicates world frame. The pull angle depends both on the humerus orientation during the wingbeat and pectoralis muscle fiber recruitment. (**C**) During downstroke, the pull angle that we compute for the doves in slow flight (black line) is lower than chukars during wing-assisted incline running (*Heers et al., 2016*) (gray line). However, if pectoralis moment is used to tension the supracoracoideus tendon during the second half of the downstroke to more finely tune the wingbeat, the pull angle would necessarily increase to values higher than plotted here and shift the curve

*Figure 6 continued on next page*

*Figure 6 continued*

towards chukar values. (**D, E**) The 3D vector direction (purple: $+x_b$: cranial, $-x_b$: ventral; green: $+y_b$: medial, $-y_b$: lateral; orange: $+z_b$: dorsal, $-z_b$: ventral) and stress magnitude (black), associated with the pectoralis pull on the humerus, were computed using the pull angle and the modeled position of the humerus under the assumption that it is the only muscle generating a moment on the wing during the downstroke.

The online version of this article includes the following figure supplement(s) for figure 6:

**Figure supplement 1.** Extra details are added to *Figure 6* to demonstrate how the 3D angular momentum balance works.

## Discussion

### Effects of angle of attack and energy storage on muscle performance

By holistically measuring the avian stroke, we provide novel insight into connections between flapping kinematics and the performance of flight muscles and tendons. Surprisingly, we find that in the region of the stroke where peak aerodynamic force occurs (*Figure 3A*), aerodynamic power dips (*Figure 4A*). This is because during mid-downstroke the doves pronate their wings to reduce the angle of attack (*Figure 2C*), which results in a local minimum in drag combined with maximal lift (*Figure 3B*), resulting in peak power factor (*Figure 3C*). Furthermore, because aerodynamic power, and hence required total muscle power, dips mid-downstroke (*Figure 4B*), the pectoralis power can be allocated to tension the supracoracoideus tendon, which simultaneously improves the pectoralis work loop shape factor (*Figure 5*). This improved shape factor flattens the dip in pectoralis power mid-downstroke (*Figure 5D*). Consequently, the work loop is more rectangular (*Figure 5E*): producing more total work at a more constant maximum power output level. Additionally, since a mid-contraction dip in power generation is suboptimal for muscle mechanics (*McMahon, 1984*), tensioning the supracoracoideus tendon is the most parsimonious interpretation of how our experimental and model outcomes can corroborate the generation of observed wing stroke kinematics. Specifically, instead of the wing continuing to accelerate mid-downstroke when velocity peaks, elastic energy storage in the supracoracoideus tendon enables the wing to reach zero acceleration midstroke, as kinematically required, after which the wing decelerates to prepare for stroke reversal. Finally, the efficient low angle of attack mid-downstroke facilitates energy storage in the supracoracoideus tendon, which is an energy-effective solution for powering the upstroke that simultaneously enables the pectoralis to generate work more efficiently—showing how the downstroke apparatus of doves benefits from being finely tuned.

The dove's mid-downstroke dip in the angle of attack (to 30.5° ± 3.1°), which has a cascading effect on drag, aerodynamic power, and energy storage, is especially pronounced compared to other species. For example, in Pacific parrotlets (*Forpus coelestis*) the mid-downstroke angle of attack is higher, roughly between 45° and 60° (*Deetjen et al., 2017*), under similar takeoff conditions for the second wingbeat. In contrast, in barn owls (*Tyto alba*) flying at higher speeds after takeoff, the angle of attack mid-downstroke is lower, roughly 30° at 5–6 m/s flight speed (*Wolf and Konrath, 2015*). A comparison with hovering birds is more representative because the advance ratio (ratio of forward flight speed to wingtip speed) of the doves during takeoff is 0.14 ± 0.01, which is close to hovering (advance ratio < 0.1; *Ellington, 1984*). For this we harness data of other specialized flying vertebrates capable of sustained or short hovering flight: hummingbirds (*Trochilidae*) and small nectar and fruit bats (*Pteropodidae* and *Phyllostomidae*) (*Ingersoll et al., 2018*). The angle of attack of the doves is most similar to that of hovering hummingbirds, whose angle of attack is 36.9° ± 4.9° (across n = 88 individuals from 17 species) during mid-downstroke, with peaks at the beginning (51.3° ± 10.9°) and end (52.7° ± 9.9°) of the downstroke (*Ingersoll et al., 2018*), so the mid-downstroke dip is flatter for hummingbirds. In contrast, the average mid-downstroke angle of attack in hovering bats is 52.2° ± 5.0° (across n = 16 individuals from two nectar and one fruit bat species, of which the latter has the highest angle of attack; *Ingersoll et al., 2018*). Notably the bat's angle of attack trace shape is more similar to that of doves. Based on our literature comparison, we find that the recorded downstroke angle of attack of doves falls between the measurements reported for other species flying at similar speeds, with the exception of the dove's pronounced dip during the mid-downstroke. Before more species are studied with high-resolution methods, we cannot attribute this pronounced dip in the angle of attack as a dove-specific specialization, hence we focus on the muscle-mechanical consequences of this wingbeat adaptation found in the dove. The immediate effect of this dip in the angle of attack is a corresponding peak in power factor. This matches our expectation based on measurements of hummingbird wings (maximum power factor of 4.35 at 27.7° angle of attack; *Kruyt*

$$x_{\mathrm{p,aero}} = f\left|\vec{\mathbf{r}}_{\mathrm{span}}\right|$$

$$x_{\mathrm{p,iner}} = \frac{m_{\mathrm{wing}}}{m_{\mathrm{body}}}\left(\frac{r_{\mathrm{gyr}}}{\left|\vec{\mathbf{r}}_{\mathrm{span}}\right|}\right)^2 \left(f\left|\vec{\mathbf{r}}_{\mathrm{span}}\right|\right)^2 f$$

**Figure 7.** We integrate body morphological and kinematic scaling laws across extant birds with our aerodynamic, inertial, and muscle power measurement for doves to predict how average muscle power and timing should scale. (**A, B**) Averaged data from the current study is plotted in red, which is scaled based on extant bird data (other colored lines; power scaled by bodyweight: bw). (**C, D**) Data from the current study is plotted in green, and scaled extant bird data is plotted in black. Data measured directly in other studies at similar flight speeds (1.23 m/s) are plotted in multiple colors and marker types, which correspond to and are explained in detail in the caption of *Figure 7—figure supplement 3*. (**A, C**) The stroke-averaged muscle power scaled by bodyweight is proportional to wingtip velocity and the aerodynamic power scaling parameter, $x_{\mathrm{p,aero}}$. Stroke-averaged pectoralis power data from the literature (fitted with dotted black line) is too scattered to confirm the trendline from our scaling analysis (solid gray line). (**B, D**) Because wing inertia dominates aerodynamic scaling, the pectoralis needs to activate earlier at larger scale (and vice versa). Hence, the timing of power and force production within the stroke scale according to the ratio of the inertial and aerodynamic power scaling parameters, $x_{\mathrm{p,iner}}/x_{\mathrm{p,aero}}$. The colored vertical lines in (**B**) are plotted at the midway point of pectoralis power exertion (*Equation S75*). This same midway point is plotted as filled dots in (**D**) with a solid gray linear-fit line. The midway point of pectoralis force exertion (*Equation S76*) is plotted as empty dots in (**D**) with a dashed gray linear-fit line. EMG timing data (colored solid lines with hashes at the start and end; starting points fitted with lower dotted black line) and corresponding peak pectoralis force data (asterisks; fitted with upper dotted black line) from the literature are too scattered to confirm the subtle trendlines from our scaling analysis.

The online version of this article includes the following figure supplement(s) for figure 7:

**Figure supplement 1.** The two overarching dynamics balance methods that we use to analyze muscle behavior resolves different information, yet are related in their form and together provide insight into the pull angle of the pectoralis on the humerus.

**Figure supplement 2.** Extra detail is added to *Figure 7*.

**Figure supplement 3.** We compare our computed stroke-averaged pectoralis power scaled by pectoralis mass with values measured in the literature using a variety of different methods, bird species, and flight speeds during level flight.

**Figure supplement 4.** By comparing our computed time-resolved power for doves (*Figure 4B*) to hummingbirds also computed using the aerodynamic force platform in *Ingersoll and Lentink, 2018*, we find that the aerodynamic power during downstroke is similar, while the aerodynamic power during upstroke and the inertial power are very different.

*et al., 2014*) and pigeon wings (maximum power factor of 6.41 at 30.2° angle of attack; *Crandell and Tobalske, 2011*). For both species, the power factor monotonically decreases at angles of attack greater than 26°. The angle of attack is hence a critical factor that has a cascading effect on down-stroke tuning: the mid-downstroke dip in the angle of attack causes a dip in drag combined with a peak in lift, resulting in peak power factor and a corresponding local minimum in aerodynamic power. Combined with the simultaneous local minimum in inertial power, this explains the local minimum in total required pectoralis power output mid-downstroke at a muscle length for which power output should ideally peak (*McMahon, 1984*).

To comprehensively understand the functional consequence of tuning the downstroke, we use our muscle model to compute bounds on dove pectoralis power with and without elastic storage in the supracoracoideus tendon. A comparison with a variety of studies across species (*Ellerby and Askew, 2007*; *Tobalske et al., 2003*; *Tobalske and Biewener, 2008*; *Biewener et al., 1998*; *Jackson and Dial, 2011a*; *Dial et al., 1997*; *Soman et al., 2005*; *Ingersoll and Lentink, 2018*; *Usherwood et al., 2005*) reveals a wide scatter of estimated pectoralis power (*Figure 7—figure supplement 3*). Some differences may be attributed to different flight modes and specie-specific adaptations, but the variance from measurement techniques also likely contributes. From the studies which vary flight speed (*Ellerby and Askew, 2007*; *Tobalske et al., 2003*), we see that flight is more expensive at low speeds (percentage increase in power at lowest measured speed compared to minimum power at any speed for zebra finch: 90.9%; budgerigar [*Melopsittacus undulatus*]: 94.9%; cockatiel [*Nymphicus hollandicus*]: 89.9%; dove: 45.2%; black-billed magpie [*Pica hudsonia*]: 142.5%) and at high speeds (percentage increase in power at highest measured speed compared to minimum power at any speed for zebra finch: 68.3%; budgeriagar: 47.5%; cockatiel: 214.1%; dove: 90.0%; magpie: 50.8%). However, this cannot explain the scatter between studies for the same species. For example, for three studies of pigeons at similar speeds (5.0 ± 0.5 m/s), the average positive pectoralis power scaled by pectoralis mass was 108 W/kg (*Biewener et al., 1998*), 207 W/kg (*Soman et al., 2005*), and 273 W/kg (*Usherwood et al., 2005*), and for three studies of magpies at similar speeds (4.2 ± 0.3 m/s), the average pectoralis power was 68 W/kg (*Tobalske et al., 2003*) (net power), 85 W/kg (*Dial et al., 1997*) (positive power), and 308 W/kg (*Jackson and Dial, 2011a*) (net power). Many of these previous studies were limited by high-variance calibrations of muscle stress (*Jackson and Dial, 2011a*), so our more sophisticated external power measurement gives valuable insight, which can be used to anchor the pectoralis power estimate for doves. The only comparable study for doves is *Tobalske et al., 2003*, which reports an average net pectoralis power of 179 W/kg for doves flying at 1 m/s. We calculated an average net pectoralis power of between 182 ± 48 W/kg (no energy storage) and 232 ± 60 W/kg (full energy storage) for doves flying at 1.23 ± 0.13 m/s, meaning that, based on our measurements, *Tobalske et al., 2003* underestimated the pectoralis power by between 1.6% ± 26.6% (no elastic storage) and 29.4% ± 33.3% (elastic storage fully powers upstroke), which is small in comparison to the variance among other studies. While we cannot, from the current measurements, determine exactly how much energy is actually stored for upstroke, we can observe that utilizing energy storage in the supracoracoideus tendon is an effective solution for helping to power the upstroke, partially because only a small increase (24.9% ± 17.5%) in pectoralis power is needed for elastic storage to obtain a well-tuned wingbeat.

A notable benefit of energy storage in the supracoracoideus tendon is that it also improves the shape factor of the pectoralis work loop. We find that for any energy storage fraction in the supracora-coideus tendon above 35%, a pectoralis work loop shape factor of over 0.72 ± 0.11 can be achieved (global maximum of 0.73 ± 0.10) if the energy storage is spread out appropriately over the course of the downstroke (*Figure 5F and G*). Compared to work loops previously reported from strain-gauge measurements of muscle stress, which are relatively triangular (data from *Tobalske et al., 2003* is plotted in *Figure 5E* for a dove in similar conditions; shape factor = 0.62; *Tobalske et al., 2003*; *Jackson and Dial, 2011a*), our more rectangular work loops suggest an improved generation of work because the relative area of positive work is greater for a rectangle than a triangle with the same maximum stress and strain.

We find additional evidence for energy storage in the supracoracoideus tendon based on the computed pull direction of the pectoralis during the downstroke (*Figure 6*). In particular, during the second half of the downstroke, the pull angle (*Figure 6C*) and pull direction (*Figure 6D and E*) of the pectoralis would more closely match previously measured kinematics in chukars (*Alectoris*

*chukar*) for wing-assisted incline running (**Heers et al., 2016**), if the supracoracoideus tendon was tensioned in that region of the wing stroke. While we should not expect the chukar kinematics to exactly match our results, as they originate from a different species and behavior (i.e., level flight *versus* wing-assisted incline running), the pull angle and direction do align well in the first half of the downstroke, which was unaffected by supracoracoideus tendon tensioning. In the second half of the downstroke, which could be affected by tensioning the tendon, the pectoralis of the chukars pulled in a caudoventral direction, whereas our measurements of the total flight muscle torque indicate a caudodorsal pull direction. Similarly, the pull angle of the pectoralis on the humerus generally agrees with the chukar kinematics in the first half of the downstroke, but diverges in the second half of the downstroke. Tensioning the tendon may explain both of these discrepancies because, firstly, the anatomy of the supracoracoideus makes it well positioned to produce a dorsal force near the end of downstroke, whereas this is impossible for the pectoralis. Secondly, if tensioning the supracoracoideus tendon increased the required pectoralis moment in the second half of the downstroke, then the pectoralis pull angle would be higher than is plotted in *Figure 6C*. While elastic tensioning of the supracoracoideus tendon seems likely (**Tobalske and Biewener, 2008**), future work measuring the supracoracoideus pull angle and direction is needed to verify these predictions. Additional future work could measure the heterogeneous recruitment of the muscle fibers in the pectoralis through the downstroke to refine the muscle moments and stresses we calculated here.

Overall, we see multifaceted evidence and benefits of extra pectoralis power production during the mid-to-late downstroke to tension the supracoracoideus tendon, which can then power much of the upstroke. In addition to reducing the work that the supracoracoideus muscle needs to produce, energy storage in the tendon enhances the pectoralis work loop shape, due to efficient aerodynamic force production mid-downstroke, caused by a dip in the angle of attack. Elastic storage also enables the pennate supracoracoideus to maximize force, for which its muscle fiber architecture is well suited, even though sonomicrometry evidence suggests it produces some work and power (**Tobalske and Biewener, 2008**). The interactions between kinematics and muscle activity are critical to understanding bird flight, hence it is valuable to study how it shifts for birds with different morphologies and flight styles.

**Table 1.** Morphological data for doves (N = 4 doves).

| Variable | Mean ± SD |
|---|---|
| Body mass (g) | 161.6 ± 11.4 |
| Wing span (tip to tip outstretched) (cm) | 50.03 ± 2.28 |
| Wing radius (tip to shoulder outstretched) (cm) | 22.58 ± 1.13 |
| Single wing area (outstretched) (cm²) | 182.6 ± 20.2 |
| Tail area (outstretched) (cm²) | 140.1 ± 17.8 |
| Aspect ratio (-) | 6.89 ± 0.48 |
| Single pectoralis mass (g) | 14.66 ± 1.97 |
| Pectoralis fascicle length (mm) | 17.12 ± 1.43 |
| Pectoralis fascicle angle (°) | 33.58 ± 2.85 |
| Pectoralis Physiological Cross-Sectional Area (PCSA) (mm²) | 690.1 ± 92.8 |
| Distance between pectoralis sonomicrometry crystals (mm) | 12.27 ± 1.70 |
| Single supracoracoideus mass (g) | 3.26 ± 0.57 |

**Table 2.** Measured kinematics and contractile properties for the second stroke after takeoff. Positive yaw angle corresponds with the dove yawing to the left. The advance ratio is forward translational velocity divided by wingtip velocity as in **Ellington, 1984** (N = 4 doves; n = 5 flights each).

| Variable | Mean ± SD |
|---|---|
| Flight speed (m/s) | 1.23 ± 0.13 |
| Downstroke length (ms) | 61.80 ± 4.73 |
| Upstroke length (ms) | 40.65 ± 3.88 |
| Flapping frequency (Hz) | 9.80 ± 0.62 |
| Yaw angle (°) | 1.62 ± 6.33 |
| Advance ratio (-) | 0.14 ± 0.01 |
| Max pectoralis strain (ε) | 0.23 ± 0.01 |
| Max pectoralis velocity (m/s) | 0.24 ± 0.04 |

## Comparative effects of wing aerodynamics and inertia in extant birds

For doves, aerodynamic power is dominant during the downstroke, and the smaller inertial power dominates during the upstroke.

In contrast, hummingbirds produce a similar amount of aerodynamic power during upstroke and downstroke (*Figure 7—figure supplement 4A*), and inertial power is dominant throughout the stroke (*Figure 7—figure supplement 4B*). The increased influence of inertia is likely due to the higher wing-beat frequency of the hummingbirds (79 Hz; *Ingersoll and Lentink, 2018*) compared to the doves (9.8 Hz; *Tables 1 and 2*). Unsurprisingly, the masses of the pectoralis and supracoracoideus muscles are more similar in hummingbirds (approximately 2:1 ratio; *Ingersoll and Lentink, 2018*) than doves (ratio of 4.60 ± 0.93) because hummingbirds need more total muscle power during the upstroke than doves (*Figure 7—figure supplement 4C*).

However, for bird species with flying styles similar to doves, the differences in muscle function and timing are more nuanced. Based on our scaling analysis, we find that average pectoralis power scaled linearly with bodyweight and wingtip speed (*Figure 7A and C*; *Equation 1*). This helps explain why flapping frequency and wingspan are inversely related across species (*Greenewalt, 1960*): to maintain a reasonable average muscle power requirement, birds with longer wings need to flap slower. The effect of inertia, on the other hand, is to impact the timing of the pectoralis power, even while the average pectoralis power is unaffected (*Figure 7B, D*; *Equation 2*). When the effects of inertia relative to aerodynamics are increased (higher wing-to-body mass ratio, wing mass more distally positioned, higher flapping frequency, longer wing length), pectoralis power needs to be produced earlier in the downstroke. Across over three orders of magnitude of extant bird sizes (between 9.5 g and 2140 g), the timing shift is small, covering a range of less than 10% of the stroke. Hence, it is difficult to determine whether previous studies measuring electrical activation timing of the pectoralis confirm this result (*Figure 7D*). However, based on our detailed mechanistic analysis, we have built new intuition for why pectoralis power (and force) should in principle be timed slightly earlier or later to tune the downstroke well. Overall, the interplay between kinematics and wing physiology is a critical factor in determining muscle function, and across a vast array of bird species with similar flight styles, the variance in timing and relative magnitude of muscle power production is small, and thus hard to measure, but functionally relevant.

## Conclusions

Our integration of *in vivo* measures of muscle activation and strain using established methods combined with novel measures of wing shape and aerodynamic force provides new insight into how the complex interplay between inertia and aerodynamics shapes wing kinematics and muscle function. While the inertial power for doves is less than the aerodynamic power, inertia plays a critical role in dictating pectoralis power production timing and dominates the required power during the upstroke. This upstroke muscle power requirement can be at least partially met by the pectoralis muscle storing elastic energy in the supracoracoideus tendon during the mid-to-late downstroke so it can be released during the upstroke. Counterintuitively, when fine-tuned correctly, this extra power production to tension the supracoracoideus tendon actually improves the effectiveness of the pectoralis muscle work loop by maximizing its work loop shape factor. Storing energy by tensioning the supracoracoideus tendon mid-downstroke effectively flattens pectoralis power production because it fills the aerodynamic power dip due to the marked reduction in the angle of attack midstroke, which improves the aerodynamic efficiency of the wing. Future studies should seek to simultaneously measure contractile behavior in the supracoracoideus (*Tobalske and Biewener, 2008*) and the strain in its tendon to provide direct tests of our predictions of elastic energy storage. This feat has not yet been performed in bird flight because the supracoracoideus tendon cannot be studied minimally invasively, underscoring the value of our model. Additionally, it would be valuable for a complete picture of whole-animal energetics to determine the ratio of muscle mechanical output to the metabolic power required for generating it (*Bundle et al., 2007*) and thus determine avian flight efficiency.

## Materials and methods

### Dove flight experiments

The experimental setup (*Figure 1—figure supplement 1*) consisted of three time-synchronized systems which imaged the 3D surface of the dove (*Figure 1C*), measured the aerodynamic forces produced by the dove (*Figure 1A and B*), and measured the activation and lengthening of the pectoralis muscles of four doves (*Figure 1A and B*). To image the 3D wing surface of each dove at 1000 Hz, we used a structured-light system (*Deetjen et al., 2017*; *Deetjen and Lentink, 2018*). We measured the vertical and horizontal aerodynamic forces produced by the dove at 2000 Hz using an AFP (*Lentink et al., 2015*; *Lentink, 2018*). Finally, sampling at 10,000 Hz, we used electromyography (EMG) to measure the electrical activation of the pectoralis (mV) and sonomicrometry to measure the strain of the pectoralis (dimensionless: change in length relative to resting length). We analyzed the second wingbeat after takeoff for four 2-year-old near-white ringneck doves (*S. risoria*; three males, one female; statistics summarized in *Table 1*, *Table 2*), which were trained to fly between perches (1.6 cm diameter) 0.65 m apart inside of the AFP. We recorded 5 flights from each dove while measuring the muscle activity for a total of 20 flights. To assess the effect of surgery and the recording cable on flight behavior, we also recorded doves 3 and 4 during five pre-surgery flights, and dove 3 during five post-surgery flights, but without the cable needed to measure muscle activity attached (*Figure 1—figure supplement 2*). The perches were mounted 0.36 m above the bottom plate of the AFP, and the residual descent angle between the takeoff and landing perch was 2°. Training involved light tapping on the tail to initiate a flight to the other perch. Some flights were rejected as outliers due to inaccurately eliciting a flight or equipment failure (noisy or missing sonomicrometry signal; insufficient suspended cable length allotment). All training, surgical procedures, and experimental procedures were approved by Stanford's Administrative Panel on Laboratory Animal Care (APLAC-30905 and APLAC-31426) and followed established methods and protocols (*Ellerby and Askew, 2007*; *Tobalske et al., 2003*; *Tobalske and Biewener, 2008*; *Biewener et al., 1998*; *Jackson and Dial, 2011a*; *Hedrick et al., 2003*; *Robertson and Biewener, 2012*).

### Muscle activation and strain measurements

To measure the electrical activation and strain of the left and right pectoralis muscles, we surgically implanted EMG electrodes and sonomicrometry crystals using standard methods for the pectoralis of birds (*Biewener et al., 1998*; *Tobalske et al., 2005*). Recordings were made by connecting a shielded cable to the back plug on the dorsal side of the dove. We loosely suspended the cable, weighing approximately 26.1 g, above the dove (*Figure 1—figure supplement 1*). Sonomicrometry signals were converted into fiber lengths, $L_{\text{pect,f}}$, by calibrating at 0, 5, and 10 mm, which we used to compute fiber strain:

$$\gamma_{\text{pect,f}} = \frac{L_{\text{pect,f}}}{L_{\text{pect,f,rest}}} - 1, \tag{4}$$

where $L_{\text{pect,f,rest}}$ is the muscle's resting length during perching. For more information, see Section A4.

### Aerodynamic force measurements

We determined the time-resolved aerodynamic force vector generated by each wing of the dove by measuring the net aerodynamic forces in the horizontal (back to front; x) and vertical (z) directions as in *Deetjen et al., 2020*; *Figure 1A and B*, *Figure 1—figure supplement 1*. We then combined these 2D forces with our 3D wing kinematics measurements to reconstruct the final, lateral (right to left), component of the full 3D force vector (*Figure 1C*). The vertical and horizontal aerodynamic forces of the dove were measured using a 2D AFP (*Lentink et al., 2015*; *Lentink, 2018*). The 2D AFP measures vertical forces by instrumenting the floor and ceiling of a flight chamber with carbon fiber composite panels. Similarly, the horizontal forces are measured with two instrumented panels on the front and back sides of the flight chamber (1 m length × 1 m height × 0.6 m width). Each of the four panels is connected in a statically determined manner to three Nano 43 sensors (six-axis, SI-9-9.125 calibration; ATI Industrial Automation) sampling at 2000 Hz with a resolution of 2 mN. We also measured takeoff and landing forces by mounting the perches on carbon fiber beams, each of which was connected in a statically determined manner to three Nano 43 sensors set on mechanically isolated support

structures. The force measurements were filtered using an eighth-order Butterworth filter with a cutoff frequency of 80 Hz for the plates and 60 Hz for the perches, or about eight and six times the flapping frequency of a dove, respectively. This enabled us to filter out noise from the setup because the natural frequencies of the force plates were all above 90 Hz and the perches had natural frequencies above 70 Hz. Validation of the setup is detailed in Section A5.

## 3D surface and kinematics measurements

We measured the 3D surface of the head, tail, and left wing of the dove, as well as multiple marker points on the dove at 1000 Hz using the same methods as described in our methods paper: *Deetjen et al., 2020*. The 3D surface of the dove (*Figure 1C*) was reconstructed using our automated structured light method (*Deetjen et al., 2017*) (five cameras and five projectors calibrated together using our automated calibration method; *Deetjen and Lentink, 2018*). We decreased the sampling speed (from 3200 Hz in *Deetjen et al., 2017*) and thickened the projected stripes to improve lighting contrast to converge on the recording setup settings described in *Deetjen et al., 2020*. Additionally, we manually tracked the following 12 feature points using triangulation: ninth primary wingtip, seventh secondary feather, shoulder, wrist, middle of the back, left and right feet, left and right eyes, tip and base of the beak, and top of the head (*Figure 1C*). For the ninth primary, shoulder, wrist, back, and the top of the head, we attached square retro-reflective marker tape and identified their centers when they were visible. The remaining positions were estimated with a combination of manual annotation and interpolation, and we smoothed the 3D reconstructed points using the 'perfect smoother' (*Eilers, 2003*) to reduce noise when taking derivatives. We combined the collected 12 kinematics (marker) points and the surface data to fit a smooth morphing surface to the body, tail, and wings of the dove. Finally, we assumed bilateral symmetry for the right wing because we focused our cameras and projectors on the left wing to maximize resolution.

## Modeling the distributed mass of the dove

We modeled the distributed mass of the wings and body of the dove as a series of point masses which move based on the tracked kinematics. For each wing, we used 20 point masses which were distributed along the wing. We fixed the location of the shoulder joint relative to the body of the dove (average of tracked shoulder positions relative to the body) and formed a V-shaped path from the shoulder joint to the wrist joint to the ninth primary feather. The point masses were then placed proportionally along this path. The masses of each point mass were determined using a scaling of the mass distribution along the wing given by Berg (*Berg and Rayner, 1995*).

## Modeling power and momentum

We developed two separate dynamics models to gain different insights into the time-resolved activity of the flight muscles. The first model is a 1D power balance of the entire dove (see Section A7 for derivation), by which we compute the time-resolved power that the dove's muscles must have generated in order to produce the kinematics and aerodynamic forces that we observed. Combined with the measured pectoralis strain rate, we computed the pectoralis force magnitude during the downstroke, assuming it is the primary muscle producing power during the downstroke. The second model is a 3D angular momentum balance of the wing (see Section A8 for derivation), by which we compute the time-resolved 3D moment vector (torque) that must have acted on the wing in order to produce the observed kinematics and aerodynamics forces. We then used the information gained from both of these models, together with skeletal measurements, to compute the time-resolved 3D force vector produced by the pectoralis during downstroke, along with its pull angle on the humerus (see Section A9 for derivation). See *Figure 7—figure supplement 1* for a side-by-side comparison and summary of the two models, and how they are used in concert to compute the pull angle of the pectoralis.

## Disentangling pectoralis and supracoracoideus power

To understand the effects that varying amounts of elastic storage in the supracoracoideus tendon would have on power generation in the pectoralis and supracoracoideus muscles, we developed a time-resolved muscle-tendon model. The output of this model is a breakdown of the time-resolved power generated, absorbed, stored, or released by the pectoralis and supracoracoideus. The sum of these four power modes across the two muscles equals the total computed muscle power at each

time step. While other muscles are involved in power generation, we simplify the model by assuming that the pectoralis and supracoracoideus generate all the necessary power. This assumption is justified because their masses significantly exceed that of the other muscles in the dove (*Tobalske and Biewener, 2008*). Using this assumption, it is clear that positive, mid-downstroke power can be attributed to the pectoralis, and positive, mid-upstroke power can be attributed to the supracoracoideus. However, two modeling challenges remain. First, during stroke reversal, *in vivo* measurements indicate that activation of the pectoralis and supracoracoideus overlaps (*Tobalske and Biewener, 2008*), adding some ambiguity in those regions. We addressed this using common-sense rules for muscle power generation: for example, muscles only generate positive power when they are shortening (see Section A10 for details). Second, the anatomy of the supracoracoideus tendon, along with *in vivo* measurements, provides evidence that it stores elastic energy during the late downstroke and releases it after stroke reversal to aid the supracoracoideus during upstroke (*Tobalske and Biewener, 2008*). Because the exact nature of this energy storage is unknown, we analyzed the effect that different amounts and timing of energy storage would have on power distribution (see Section A11 for details).

## Scaling analysis across extant birds

We scaled our aerodynamic, inertial, and muscle measurements for doves across multiple extant birds to estimate comparative patterns of power output during slow flight. Assuming that aerodynamic forces scale with bodyweight, wing velocity scales with flapping frequency times wingspan, and using our derived modeling equations, we traced the effects of aerodynamics and inertia to formulate key scaling parameters that dictate muscle performance. In particular, the aerodynamic power is proportional to the wingtip velocity (note the force associated with aerodynamic power is proportional to bodyweight, we thus do not dissect its dependence on velocity squared explicitly), and the inertial power is proportional to the wing mass ratio (ratio of wing mass to bodyweight), the wing radius of gyration ratio (ratio of radius of gyration to wingspan) squared, the wing length, and the wingbeat frequency. For more details, see Section A12.

## Acknowledgements

We thank Wren Cooperrider for his help acquiring data for this study and Paul Mitiguy for his consultation during algorithm development related to dynamics. This work was supported by an NSF Faculty Early Career Development (CAREER) Award 1552419 to DL and motivated by NSF reviewer feedback. Additional support to study wing morphing came from AFOSR BRI award number FA9550-16-1-0182, with special thanks to BL Lee for leading the program. BWT was funded by the National Science Foundation grant NSF EFRI 1935216, MED was supported by a Stanford University Graduate Fellowship, and DDC was supported by a Stanford Graduate Fellowship and a National Defense Science and Engineering Graduate Fellowship.

## Additional information

### Competing interests
David Lentink: Reviewing editor, eLife. The other authors declare that no competing interests exist.

### Funding

| Funder | Grant reference number | Author |
| --- | --- | --- |
| National Science Foundation | Graduate Research Fellowship Program DGE-114747 | Marc E Deetjen |
| Air Force Office of Scientific Research | FA9550-16-1-0182 | David Lentink |
| National Science Foundation | CAREER Award 1552419 | David Lentink |

| Funder | Grant reference number | Author |
|---|---|---|
| Stanford University | Stanford Graduate Fellowship | Marc E Deetjen Diana D Chin |
| National Defense Science and Engineering Graduate | | Diana D Chin |
| National Science Foundation | IOS-1838688 | Bret W Tobalske |
| National Science Foundation | EFRI 1935216 | Bret W Tobalske |

The funders had no role in study design, data collection and interpretation, or the decision to submit the work for publication.

## Author contributions

Marc E Deetjen, Conceptualization, Resources, Data curation, Software, Formal analysis, Validation, Investigation, Visualization, Methodology, Writing – original draft, Project administration, Writing – review and editing, Contributed to collecting and analyzing data from the dove experiments, interpreting the findings, developing all new mathematical models and derivations, and drafting the manuscript; Diana D Chin, Resources, Data curation, Software, Validation, Methodology, Writing – review and editing, Contributed to collecting and analyzing data from the dove experiments; Ashley M Heers, Resources, Software, Validation, Writing – review and editing, Contributed to developing the musculoskeletal model; Bret W Tobalske, Conceptualization, Resources, Data curation, Validation, Investigation, Methodology, Writing – review and editing, Contributed to collecting and analyzing data from the dove experiments, conducting the surgical procedure, and editing the manuscript; David Lentink, Conceptualization, Resources, Data curation, Supervision, Funding acquisition, Investigation, Methodology, Project administration, Writing – review and editing, Contributed to collecting and analyzing data from the dove experiments, interpreting the findings, and editing the manuscript. DL also oversaw the project

## Author ORCIDs

Marc E Deetjen ⬡ https://orcid.org/0000-0002-6947-6408
Diana D Chin ⬡ http://orcid.org/0000-0002-3015-7645
Ashley M Heers ⬡ http://orcid.org/0000-0002-7635-0651
Bret W Tobalske ⬡ https://orcid.org/0000-0002-5739-6099
David Lentink ⬡ https://orcid.org/0000-0003-4717-6815

## Ethics

All training, surgical procedures, and experimental procedures were approved by Stanford's Administrative Panel on Laboratory Animal Care (APLAC-30905 and APLAC-31426). Details and citations provided in the Materials and Methods section.

Reviewer #1 (Public Review): https://doi.org/10.7554/eLife.89968.3.sa1
Reviewer #2 (Public Review): https://doi.org/10.7554/eLife.89968.3.sa2
Author Response https://doi.org/10.7554/eLife.89968.3.sa3

# Additional files

## Supplementary files

• Supplementary file 1. Measured aerodynamic property tables. (a) Air conditions across all 20 flights. N = 4 doves; n = 5 flights each. (b) Measured aerodynamic and leg impulses before surgery, after surgery, and after surgery with the suspended recording cable (for recording EMG and sonomicrometry). So that horizontal and vertical impulses can be compared, gravity has been subtracted from the vertical impulse, so that while at rest, the vertical impulse is zero (horizontal and vertical impulse scaled by bodyweight, bw and integration time; bodyweight is based on the average vertical force on the takeoff perch before takeoff). These definitions for impulse match *Chin and Lentink, 2017* paper on directional impulse of birds (*Chin and Lentink, 2017*). (c) Measured aerodynamic and leg impulses among individual doves. The definition for impulse

matches Supplementary file 1b. (d) Measured aerodynamic forces for the second stroke after takeoff (*Figure 3*). The weight of the dove is primarily supported during the downstroke. Drag partially contributes to weight support, while opposing forward aerodynamic force. The reported values are the stroke-averaged aerodynamic force measured by the aerodynamic force platform (AFP), normalized by bodyweight. We compute the time-resolved lift and drag using the measured horizontal and vertical aerodynamic forces together with the measured wing velocity. N = 4 doves; n = 5 flights each; mean ± standard deviation. (e) We compute aerodynamic, inertial, and required muscle power based on first principals (*Figure 4*) and model the effect of the amount of energy storage in the supracoracoideus (*Figure 5*). The primary component of stroke-averaged net muscle power is aerodynamic, whereas the positive and negative regions of inertial power cancel. Our energy storage model predicts that the pectoralis would need to generate 24.9% ± 17.5% more power to fully power the upstroke via energy storage in the supracoracoideus. N = 4 doves; n = 5 flights each; mean ± standard deviation.

• Supplementary file 2. Scaling analysis (*Figure 7*) parameter summary. (a) Scaling analysis (*Figure 7*) parameter summary. (b) Scaling analysis parameters used for analyzing shifts in the timing of the pectoralis power distribution during the stoke (*Figure 7D*).

• Supplementary file 3. Detailed breakdown of data from extant birds used for the scaling analysis (*Figure 7*). (a) Detailed breakdown of data from extant birds used for the scaling analysis (*Figure 7*). Some data was scaled isometrically by bodyweight based on matching the species of the bird (blue). Other data was found based on published scaling relationships (green). The references corresponding to the letter plus number keys (found in columns titled Src) are in Supplementary file 3c. To match some of the bird species, a close relative was used when data was missing for the original species: close relative of *Parus ater* used was *Parus cristatus mitratus*. Close relative of *Perisoreus canadensis* used was *Xanthoura yncas*. Close relative of *Pica hudsonia* used was *Pica pica*. Close relative of *Corvus brachyrhynchos* used was *Corvus vorone*. Close relative of *Alectoris rufa* used was *Coturnix*. Close relative of *Corvus corax* used was *Corvus vornix*. (b) Detailed breakdown of data from extant birds used for analyzing shifts in the timing of the pectoralis power distribution during the stoke for the scaling analysis (*Figure 7D*). The same color scheme is used as in Supplementary file 3a, and the close relative species used when data was missing for the original species are: close relative of *Selasphorus rufus* used was *Calypte anna*. Close relative of *Perisoreus canadensis* used was *Xanthoura yncas*. Close relative of *Pica hudsonia* used was *Pica pica*. Close relative of *Corvus brachyrhynchos* used was *Corvus corone*. Close relative of *Alectoris chukar* used was *Caccabis rufa*. Close relative of *Corvus corax* used was *Corvus cornix*. (c) Keys for references in Supplementary files 3a and b.

• MDAR checklist

### Data availability

Raw data, processed data, and software are available from the Dryad Digital Repository: https://doi.org/10.5061/dryad.wwpzgmsqs.

The following dataset was generated:

| Author(s) | Year | Dataset title | Dataset URL | Database and Identifier |
| --- | --- | --- | --- | --- |
| Marc D, Diana C, Ashley H, Bret T, David L | 2023 | How small deviations in kinematics and body form dictate muscle performances in the finely tuned avian downstroke | https://doi.org/10.5061/dryad.wwpzgmsqs | Dryad Digital Repository, 10.5061/dryad.wwpzgmsqs |

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

## Appendix 1

### Supplementary results

#### Section A1: Effects of surgery

While many studies involving surgical procedures have been conducted to study the pectoralis (**Biewener et al., 1992**; **Dial and Biewener, 1993**; **Biewener et al., 1998**; **Hedrick et al., 2003**; **Tobalske et al., 2003**; **Ellerby and Askew, 2007**; **Tobalske and Biewener, 2008**; **Jackson and Dial, 2011a**; **Robertson and Biewener, 2012**), few quantified the change in flight performance due to the surgery (**Tobalske et al., 2005**). While we primarily analyzed the second wingbeat that the doves made after takeoff after undergoing surgery, we recognize that this may constrain our understanding of dove flight in general. Hence, we first analyzed the effects of the surgical procedure and muscle recording cable on the overall flight dynamics for one dove for which we made recordings before as well as after the surgical procedures.

For this one dove, surgery consistently changed the way it took off and flew (**Figure 1—figure supplement 2**). After the surgery, the dove pushed off from the takeoff perch with similar vertical force (vertical impulse before: 0.39±0.03; after: 0.34±0.03), but substantially less forward force (horizontal impulse before: 0.81±0.03; after: 0.56±0.05; gravity was subtracted from vertical impulses; all impulses were scaled by bodyweight and integration time). During flight, the dove produced a smaller braking force after the surgery (horizontal impulse before: –0.09±0.02; after: –0.03±0.01). Additionally, the dove used more wingbeats to fly the same distance, resulting in a higher flapping frequency (before: 8.28±0.19 Hz; after: 9.20±0.15 Hz), and a different first stroke. The first downstroke was smaller in magnitude (maximum force scaled by bodyweight before: 2.18±0.13; after: 1.09±0.05) and began 14.4±4.6 milliseconds before release from the perch rather than 54.9±12.7 milliseconds after release from the perch.

The effects of the suspended recording cable were less pronounced than the effects of surgery, although still noticeable. The mass of the entire cable was 26.1 grams, but the extra mass moved by the dove was only an average of 10.9±3.8 grams (6.4% ± 2.2% of bodyweight) based on bodyweight measurements recorded while the dove was sitting on the takeoff perch. When the dove was carrying the extra mass of the cable, it pushed off from the takeoff perch in a more forward direction (vertical impulse scaled by bodyweight before: 0.34±0.03; after scaled by bodyweight plus cable: 0.29±0.05; horizontal impulse scaled by bodyweight before: 0.56±0.05; after scaled by bodyweight plus cable: 0.64±0.03). Once airborne, the dove produced greater vertical impulse (before: –0.28±0.04; after: –0.23±0.06), and greater forward impulse (before: –0.03±0.01; after: 0.02±0.03), while flapping more frequently (before: 9.20±0.15 Hz; after: 9.62±0.22 Hz). However, all of these flight differences are small compared to the differences caused by the surgery.

When we compare the aerodynamic forces produced before and after surgery by dove three (**Figure 1—figure supplement 2** and **Supplementary file 1b**), we find an increase in flapping frequency after surgery which mirrors the differences in flight performance reported for a small passerine (**Tobalske et al., 2005**). Additionally, after surgery, we consistently observed a modified takeoff pattern: the dove pushed off with less forward force, and its first downstroke was earlier and produced less aerodynamic force. Overall, the flight performance of the dove was substantially altered by the surgery, and to a lesser degree, by the weight increase due to the attached recording cable. However, these changes are analogous to changing the task the dove needs to accomplish, similar to changing flight distance, speed, or angle. Our results therefore quantify this particular, post-surgery flight condition, rather than flight in general.

#### Section A2: Flight styles among individuals

The four doves used somewhat different flight strategies from one another (**Figure 2B**, **Figure 1—figure supplement 3**), yet parameters during the second downstroke were similar across 20 flights (**Figures 1A and 2**, **Supplementary file 1c**). During takeoff, dove 1 pushed off with the most force, while dove 4 pushed off with the least force in both the vertical (vertical impulse for dove 1: 0.34±0.04; dove 4: 0.23±0.06) and horizontal directions (horizontal impulse for dove 1: 0.77±0.06; dove 4: 0.44±0.04; gravity was subtracted from vertical impulses; impulses herein are dimensionless as they were scaled by bodyweight (N) and integration time (s)). Because dove 1 pushed off with greater forward force, it could start braking earlier in the flight than dove 4, which still needed to produce forward thrust during the second stroke after takeoff (horizontal impulse for dove 1: –0.09±0.01;

dove 4: 0.17±0.03). This may explain the difference in stroke plane angle during the second downstroke for the two doves (*Figure 2B*; dove 1: 31.6°±2.5°; dove 4: 45.07°±2.8°), even though there is not substantial variation in the other kinematic angles among the doves (*Figure 2C*): whereas the aerodynamic force magnitude needed was similar for both doves due to weight support, the force direction was different. The vertical force during the second stroke was additionally influenced by the timing and number of strokes the doves took (number of strokes between takeoff and landing for dove 1: 4; dove 2: 5; dove 3: 4; dove 4: 5), but, overall, we found that the vertical aerodynamic force was relatively consistent during the second stroke (vertical impulse for all doves: 0.86±0.09; gravity not subtracted here for clarity).

During the second stroke, the pectoralis muscle performance was consistent across the 20 flights (*Figure 1A*, *Figure 1—figure supplement 3*). The electrical activation of the pectoralis began during the middle portion of the upstroke, 7.07±5.10 milliseconds before the pectoralis began shortening, which occurred 10.54±2.86 milliseconds before the beginning of the downstroke (defined by kinematic stroke reversal: see stroke angle in *Figure 2C*). The electrical activation then lasted 37.63±7.66 milliseconds and ended 68.57±3.38 milliseconds before the pectoralis completed contracting, which occurred 11.06±1.99 milliseconds before the end of the downstroke. The pectoralis shortened for 59.62±3.84 milliseconds during the stroke which lasted 102.45±6.59 milliseconds (downstroke: 61.80±4.73 milliseconds). Furthermore, we see that the timing of the pectoralis contraction corresponds with the majority of aerodynamic force production.

## Section A3: Analysis of 3D muscle torques

Using the 3D angular momentum balance model (*Figure 6*), we can compute the time-resolved moment vector (torques) generated by the pectoralis muscle during the downstroke. Due to the complexity of 3D moments, we consider each of the three components of muscle torque on the wing separately.

First, we consider the moment in the roll axis ($x_b$ points in the cranial direction and is aligned with the thoracic vertebrae of the dove; *Figure 6B*, *Figure 6—figure supplement 1B*). During the majority of the downstroke, the flight muscles pulled ventrally on the wing (average ventral moment during downstroke: 0.140±0.013 N-m) to oppose lift and drag, whose world z-axis components each point vertically up (*Figure 3—figure supplement 1C*). Just before the end of the downstroke, and 1.97±1.91 milliseconds before the pectoralis began lengthening, inertial effects from decelerating the wing were associated with a small dorsal moment (average dorsal moment at the end of the downstroke: 0.057±0.026 N-m). As is the case for power, inertia dominated during the upstroke.

Second, we consider the moment in the pitch axis ($y_b$ points laterally towards the distal wing, perpendicular to the thoracic vertebrae; *Figure 6—figure supplement 1C*), in which the effects of aerodynamics dominated over inertia. When we examine the lift and drag forces (*Figure 3*, *Figure 3—figure supplement 1*), we again see that the vertical direction was the most important during the downstroke because forces in the lateral direction created no $y_b$ moment, and the horizontal forces were small (15.5% ± 4.1% of the stroke-averaged net aerodynamic force during the downstroke). At the beginning of the downstroke, the wing was at the posterior of the bird, creating a positive aerodynamic torque in the $y_b$ axis, which was balanced by a negative muscle moment in the $y_b$ axis (flight muscle acting to supinate the wing). At the end of the downstroke, the wing was at the anterior of the bird, with a negative aerodynamic torque. Hence, the total muscle moment in the pitching axis acted to first supinate (average supination moment during beginning of downstroke: 0.047±0.010 N-m), then pronate (average pronation moment during end of downstroke: 0.068±0.026 N-m), the wing during the downstroke. Since the pectoralis is thought mainly to pronate the wing, it appears that other flight muscles are driving the small supination moment at the beginning of the downstroke.

Finally, we consider the moment in the yaw axis ($z_b$ points in the dorsal direction; *Figure 6—figure supplement 1D*), where aerodynamics again dominate the downstroke. To gain a more holistic view of the relationship between the moment in the yaw axis and the horizontal, lateral, and vertical world components of lift and drag, it is helpful to examine *Figure 6—figure supplement 1E–G*, where we plot the 3D moments in the world coordinate system. The aerodynamic moment in the vertical world axis was small throughout the downstroke (average magnitude: 0.029±0.009 N-m), because the vertical lift and drag create no moment in the vertical direction (*Figure 6—figure supplement 1G*). Therefore, the yaw aerodynamic moment originated primarily from the moment in

the horizontal world axis, rotated about the lateral world axis by the amount that the dove pitches its body forward. The net effect was to generate an average muscle moment of 0.063±0.017 N-m in the caudal direction during the downstroke.

As is the case for the power analysis, the effects of the aerodynamics dominated during the downstroke, whereas inertia dominates during the upstroke and near stroke reversal. Furthermore, for all three components of the aerodynamic moment, we find that the components of lift and drag in the vertical world direction were dominant.

## Appendix 2

### Supplementary methods

#### Section A4: Muscle activation and strain measurement details

To measure the electrical activation and strain of the left and right pectoralis muscles, we surgically implanted EMG electrodes and sonomicrometry crystals using standard methods for the pectoralis of birds (**Biewener et al., 1998**; **Tobalske et al., 2005**). The doves were anesthetized using isoflurane by inhalation. Then, each pair of sonomicrometry crystals (2.0 mm; Sonometrics, Inc, London, ON, Canada) and each custom-made, fine-wire bipolar EMG hook electrode (0.5 mm bared tips with 2 mm spacing; California Fine Wire, Inc, Grover Beach, CA, USA) were implanted parallel to the fascicle axis of the mid-anterior region of sternobrachial portion of pectoralis at a depth of approximately 4 mm beneath the superficial fascia of the muscle. Before suturing (4–0 coated Vicryl) closed all of the incisions, the sonomicrometry signals were tested to ensure good signal quality. Then, all of the electrodes were sutured to superficial connective tissue a few millimeters away from the exit point of the pectoralis, to prevent movement independent of the fascicles. We left a small loop in the EMG electrode at the surface of the muscle to limit low-frequency noise. The electrode and transducer wires were tunneled subscutaneously along a narrow path from the ventral incision, along the lateral side of the dove, to the dorsal surface where the wires terminated into a back plug. The back plug was secured to the dove's back by suturing the base of the plug to intervertebral ligaments, and it was custom-made (prior to surgery) using minature connectors and epoxy. The skin was sutured closed around the back plug. After the experiments, the doves were euthanized using an overdose of isoflurane. All training, surgical procedures, and experimental procedures were approved by Stanford's Administrative Panel on Laboratory Animal Care (APLAC-30905 and APLAC-31426) and followed established methods and protocols (**Biewener et al., 1998**; **Heers et al., 2016**; **Tobalske et al., 2003**; **Ellerby and Askew, 2007**; **Tobalske and Biewener, 2008**; **Jackson and Dial, 2011a**; **Robertson and Biewener, 2012**).

Recordings were made by connecting a shielded cable (flexible wires sheathed approximately every 30 cm; diameter 7 mm) to the back plug on the dorsal side of the dove. We loosely suspended the cable above the dove (**Figure 1—figure supplement 1**), and the length of the cable that was suspended was approximately 76 mm, weighing 26.1 g. The cable was connected to an Ultrasound Dimension Gauge (UDG; Sonometrics, Inc, London, ON, Canada) to measure the sonomicrometry signals, and to a differential AC amplifier (Model 1700; A-M Systems, Sequim, WA, USA) to measure the EMG signals. All signals were recorded at 10,000 Hz using a Digidata 1550 A A/D converter (Axon Instruments, Union City, CA, USA). Sonomicrometry signals were converted into fiber lengths, $L_{\text{pect,f}}$, by calibrating at 0 mm, 5 mm, and 10 mm, and assuming the speed of sound transmission is 1590 m·s$^{-1}$ when the muscle is at 37 degrees Celsius (**Marsh, 2016**). Then, these fiber lengths were converted into fiber strain, $\gamma_{\text{pect,f}}$, using **Equation 4**. We corrected for a time delay of 2ms (2.0% of the stroke) based on the time delay in the UDG, whereas the EMG signals did not have any time lag. To compare the EMG signals across all flights, we filtered the rectified signal with a second order butterworth filter with a cutoff frequency of 50 Hz. While we recorded EMG signals for both the left and right pectoralis muscles, the right signal was unreliable, so we only include data from the left EMG.

#### Section A5: Aerodynamic force measurement validation

The 2D AFP setup (**Lentink, 2018**; **Lentink et al., 2015**) was validated by tethering a quadcopter to an instrumented beam and then comparing vertical and horizontal forces measured by the beam with forces measured by the AFP, as reported in (**Lentink et al., 2015**). Over a period of 10 seconds and a sampling rate of 1000 Hz, the impulse ratio and the mean force ratio of the 2D AFP to the beam were both 1.00±0.02 (n=10 trials) in the vertical direction and 1.00±0.01 in the horizontal direction (n=20 trials). Additionally, for a flight that starts and ends at rest, we expect that the total vertical impulse imparted by the legs and wings should equal full bodyweight support (**Chin and Lentink, 2017**), and that the net horizontal impulse should equal zero. Integrating the forces from takeoff to landing for the dove's flight, we measured a vertical impulse ratio (impulse from legs and wings divided by impulse due to bodyweight, bw) of 1.03±0.02, and a horizontal impulse (impulse from legs and wings divided by bodyweight) of 0.02±0.02 bw·s for 15 flights where the doves did not have an external cable attached to the back. For the 20 flights where the doves had a cable attached, we measured a vertical impulse ratio of 1.01±0.02, and a horizontal impulse of 0.08±0.02 bw·s. The horizontal impulse is greater than zero when the cable is attached, because the dove did positive work to pull the cable forward.

## Section A6: Aggregate aerodynamic and kinematic parameters

To compare all flights and understand the underlying dynamics, we reduced the highly resolved 3D surface and kinematic measurements, in combination with the measured aerodynamic forces, into simplified aerodynamic and kinematic parameters which summarize the time-resolved state of entire wing.

To facilitate this simplification, we divided the wing into blade elements spaced 1 mm apart in the spanwise direction. For each blade element (J total blade elements, index j), we computed: (1) the surface area, $S_j$, (2) the chordline vector from the trailing edge to the leading edge, (3) the velocity vector at the quarter-chord location (25% chord length behind the leading edge; measured relative to the Newtonian reference frame, N), $^N\vec{\mathbf{v}}^j$, which includes the velocity components of both the flapping and morphing motion of the wing, and (4) the angle of attack induced by wing motion, $\alpha_j$, which is the angle between the chordline and velocity vectors (**Figure 2A and C**).

We computed many of the simplified wing parameters using the spanwise blade element measurements and the 3D kinematics measurements. From the spanwise blade element measurements, we computed representative wing measurements which are the weighted averages of the blade element values (derived in **Deetjen et al., 2020**):

$$\vec{\mathbf{v}}_{\text{wing}} = \frac{\sum_{j=1}^{J} S_j \left| ^N\vec{\mathbf{v}}^j \right|^2 {}^N\vec{\mathbf{v}}^j}{\sum_{j=1}^{J} S_j \left| ^N\vec{\mathbf{v}}^j \right|^2} , \tag{S1}$$

$$\alpha_{\text{wing}} = \frac{\sum_{j=1}^{J} S_j \left| ^N\vec{\mathbf{v}}^j \right|^2 \alpha_j}{\sum_{j=1}^{J} S_j \left| ^N\vec{\mathbf{v}}^j \right|^2} . \tag{S2}$$

The weight for each blade element is the surface area times the velocity magnitude squared. To precisely track the position and orientation of the wing, we defined the stroke plane angle, $\phi_{\text{stroke}}$, stroke angle, $\theta_{\text{stroke}}$, deviation angle, $\theta_{\text{deviation}}$, and twist angle, $\theta_{\text{twist}}$, of the wing (**Figure 2A**). The stroke plane was linearly fit using the path swept by the tip of the ninth primary feather across the full wingbeat, where a positive stroke plane angle corresponds to a stroke plane which is pitched down. The stroke angle and deviation angle are defined relative to the stroke plane according to **Figure 2A**, and the twist angle is the angle between the chordline vector at the root and the chordline vector along the span of the wing. Finally, we also computed time-resolved parameters to describe wing morphing. The semi-wing surface area, $S_{\text{wing}}$, is the area of a single wing:

$$S_{\text{wing}} = \sum_{j=1}^{J} S_j, \tag{S3}$$

which we used to calculate the aspect ratio, $AR_{\text{wing}}$, of the wing:

$$AR_{\text{wing}} = \frac{b_{\text{wing}}^2}{S_{\text{wing}}}, \tag{S4}$$

where the wing radius, $b_{\text{wing}}$, is the length of the V-shaped path from the shoulder joint to the wrist joint to the ninth primary feather. The folding ratio, $FR_{\text{wing}}$, of the wing:

$$FR_{\text{wing}} = \frac{\left| \vec{\mathbf{r}}_{\text{span}} \right|}{\max \left| \vec{\mathbf{r}}_{\text{span}} \right|}, \tag{S5}$$

depends on the span vector, $\vec{\mathbf{r}}_{\text{span}}$, which is the vector from the shoulder joint to the ninth primary feather. The max function finds the maximum value across the stroke.

To compute the lift and drag forces and the lateral (right to left) aerodynamic forces produced by the wing (**Figure 1C**), we utilized the measured vertical and horizontal aerodynamic forces, in

addition to the 3D surface and kinematic measurements. From the derivation in **Hedrick et al., 2003** (**Deetjen et al., 2020**), the directions of lift, $\hat{\mathbf{L}}_L$, and drag, $\hat{\mathbf{D}}_L$, for the entire left wing are:

$$\hat{\mathbf{D}}_L = -\frac{\sum_{j=1}^{n} S_j \left| {}^N\vec{\mathbf{v}}^j \right| {}^N\vec{\mathbf{v}}^j}{\left| \sum_{j=1}^{n} S_j \left| {}^N\vec{\mathbf{v}}^j \right| {}^N\vec{\mathbf{v}}^j \right|}, \tag{S6}$$

$$\hat{\mathbf{L}}_L = \pm\frac{\hat{\mathbf{D}}_L \times \vec{\mathbf{r}}_{span}}{\left| \hat{\mathbf{D}}_L \times \vec{\mathbf{r}}_{span} \right|} \tag{S7}$$

To compute the magnitude of lift, $L$, and drag, $D$, on both wings, we included the net measured horizontal, $F_{AFP,x}$, and vertical, $F_{AFP,z}$, aerodynamic forces:

$$D = \frac{\left( \hat{L}_{L,x}\cos^2\theta - \hat{L}_{L,y}\cos\theta\sin\theta \right) F_{AFP,z} - \hat{L}_{L,z}F_{AFP,x}}{2\bar{S}}, \tag{S8}$$

$$L = -\frac{\left( \hat{D}_{L,x}\cos^2\theta - \hat{D}_{L,y}\cos\theta\sin\theta \right) F_{AFP,z} - \hat{D}_{L,z}F_{AFP,x}}{2\bar{S}}, \tag{S9}$$

$$\bar{S} = \left( \hat{L}_{L,x}\cos^2\theta - \hat{L}_{L,y}\cos\theta\sin\theta \right) \hat{D}_{L,z} - \hat{L}_{L,z} \left( \hat{D}_{L,x}\cos^2\theta - \hat{D}_{L,y}\cos\theta\sin\theta \right), \tag{S10}$$

where the x and z subscripts indicate the components of the lift and drag unit vectors, and $\theta$ is the yaw angle of the dove. A positive yaw angle corresponds with the dove yawing to the left. Finally, we summed the lift and drag vectors to compute the total 3D aerodynamic force vector produced by the wing, $\vec{\mathbf{F}}_{wing}$:

$$\vec{\mathbf{F}}_{wing} = D\hat{\mathbf{D}} + L\hat{\mathbf{L}}. \tag{S11}$$

Near stroke reversal, these equations encounter a singularity, so we smoothed the angle of attack, and lift and drag forces in these regions using the `perfect smoother' (**Eilers, 2003**) with smoothing weights, $W$, based on the same approach as in **Deetjen et al., 2020**:

$$W = 1 - max\left\{ 0, min\left\{ 1, \frac{log|\bar{S}| - logc_1}{logc_0 - logc_1} \right\} \right\}, \tag{S12}$$

with $c_0 = 0.15$ and $c_1 = 0.35$.

## Section A7: 1D power balance of the entire dove
To compute the time-resolved power that the dove's muscles must have produced, we balanced the muscle power with the effects of the aerodynamic forces and inertia.

To do this, we first needed to decide what the most effective scope to analyze power is: power balance of a single wing of the dove, or power balance of the entire dove. To aid in this decision, we examined the definition of power, ${}^NP^B$, for a system B (a collection of point masses and rigid bodies: scope of the power analysis) in the Newtonian reference frame N:

$$^NP^B = \frac{d^NK^B}{dt}, \tag{S13}$$

where ${}^NK^B$ is the kinetic energy of system B in reference frame N. Specifying what system B encompasses was needed to clarify how terms relating to the muscles, aerodynamics, and inertia would be incorporated into this high-level equation. One possible way to define system B was as a system that contains only point masses and rigid bodies which model a single wing of the dove. This definition is advantageous because it eliminates the term related to the kinetic energy of the body, which involves the second derivative of the position of the body. However, the disadvantage of this definition is that an extra power term appeared which we could not measure. That is, we had no

method to measure the reaction force on the shoulder joint, which contributed to power because it was an external force acting on the wing. The only alternative way to eliminate this term was to assume that the shoulder joint itself moved in a Newtonian reference frame (constant velocity and no rotation), which is a large over-simplification. Instead, we defined system B to contain the entire dove, and accepted the difficulty in computing the second derivative of the position of the body, which could add noise to the results. With the scope of the power balance decided, we expanded each side of *Equation S13* to incorporate the effects of the muscles, aerodynamics, and inertia.

The kinetic energy on the right side of *Equation S13* is made up of the distributed masses of the wings and body. This term expands to:

$$^{\mathrm{N}}K^{\mathrm{B}} = \frac{1}{2}\sum_{\mathrm{i}=1}^{\mathrm{I_{W,B}}}\left(m_{\mathrm{i}}{}^{\mathrm{N}}\vec{v}^{\mathrm{i}}\cdot{}^{\mathrm{N}}\vec{v}^{\mathrm{i}} + {}^{\mathrm{N}}\vec{\omega}^{\mathrm{B_i}}\cdot\vec{\mathbf{I}}^{\mathrm{B_i}}\cdot{}^{\mathrm{N}}\vec{\omega}^{\mathrm{B_i}}\right),$$ (S14)

where $m_{\mathrm{i}}$ is the mass of the i$^{\mathrm{th}}$ of $\mathbf{I}_{\mathrm{W,B}}$ point masses in the body and wings, $^{\mathrm{N}}\vec{v}^{\mathrm{i}}$ is the velocity vector of the center of mass of the i$^{\mathrm{th}}$ point mass in the Newtonian reference frame, $^{\mathrm{N}}\vec{\omega}^{\mathrm{B_i}}$ is the angular velocity of the reference frame of the i$^{\mathrm{th}}$ body in system B relative to the Newtonian reference frame, and $\vec{\mathbf{I}}^{\mathrm{B_i}}$ is the moment of inertia of the i$^{\mathrm{th}}$ body in system B relative to the center of mass of that body. Since we modeled the dove using point masses, *Equation S14* simplifies to:

$$^{\mathrm{N}}K^{\mathrm{B}} = \frac{1}{2}\sum_{\mathrm{i}=1}^{\mathrm{I_{W,B}}}m_{\mathrm{i}}{}^{\mathrm{N}}\vec{v}^{\mathrm{i}}\cdot{}^{\mathrm{N}}\vec{v}^{\mathrm{i}}$$ (S15)

The power term on the left side of *Equation S13* is made up of the external power exerted on the dove and the internal power produced by the dove. Assuming that all the internal power produced by the dove was accounted for by the two pectorales and supracoracoideuses, the dove's power expands to:

$$^{\mathrm{N}}P^{\mathrm{B}} = 2{}^{\mathrm{N}}P^{\mathrm{pect}} + 2{}^{\mathrm{N}}P^{\mathrm{supra}} - {}^{\mathrm{N}}P^{\mathrm{aero}} - {}^{\mathrm{N}}P^{\mathrm{g}},$$ (S16)

where $-{}^{\mathrm{N}}P^{\mathrm{aero}}$ is the power done by external aerodynamic forces, $-{}^{\mathrm{N}}P^{\mathrm{g}}$ is the power done by external gravitational forces, $^{\mathrm{N}}P^{\mathrm{pect}}$ is the internal power produced by each pectoralis muscle, and $^{\mathrm{N}}P^{\mathrm{supra}}$ is the internal power produced by each supracoracoideus muscle. We derived the aerodynamic power in *Deetjen et al., 2020*:

$$^{\mathrm{N}}P^{\mathrm{aero}} = -2\vec{\mathbf{F}}_{\mathrm{aero}}\cdot\vec{\mathbf{v}}_{\mathrm{wing}},$$ (S17)

where $\vec{\mathbf{F}}_{\mathrm{aero}}$ is the overall aerodynamic force on the wing, and $\vec{\mathbf{v}}_{\mathrm{wing}}$ is the representative wing velocity vector computed using , *Equation S1* . The gravitational power is the summation of the gravity forces on each point mass dotted with their velocity vectors:

$$^{\mathrm{N}}P^{\mathrm{g}} = -\sum_{\mathrm{i}=1}^{\mathrm{I_{W,B}}}m_{\mathrm{i}}\vec{\mathbf{g}}\cdot{}^{\mathrm{N}}\vec{\mathbf{v}}^{\mathrm{i}},$$ (S18)

where $\vec{\mathbf{g}}$ is the gravity vector. This equation can be simplified, showing that only the velocity in the z direction is relevant for gravitational power:

$$^{\mathrm{N}}P^{\mathrm{g}} = \sum_{\mathrm{i}=1}^{\mathrm{I_{W,B}}}m_{\mathrm{i}}g{}^{\mathrm{N}}v_{\mathrm{z}}^{\mathrm{i}},$$ (S19)

where $g$ is the magnitude of gravity and $^{\mathrm{N}}v_{\mathrm{z}}^{\mathrm{i}}$ is the vertical component of the velocity vector.

To compare the effects on required muscle power from terms related to inertia *versus* aerodynamics, we grouped all of the terms related to inertia together, and named the resulting term inertial power:

$$^{\mathrm{N}}P^{\mathrm{inertia}} = \frac{d{}^{\mathrm{N}}K^{\mathrm{B}}}{dt} + {}^{\mathrm{N}}P^{\mathrm{g}},$$ (S20)

$$^{\mathrm{N}}P^{\mathrm{inertia}} = \frac{1}{2}\frac{d}{dt}\left(\sum_{i=1}^{I_{\mathrm{W,B}}} m_i{}^{\mathrm{N}}\vec{\mathbf{v}}^i \cdot {}^{\mathrm{N}}\vec{\mathbf{v}}^i\right) + \sum_{i=1}^{I_{\mathrm{W,B}}} m_i g{}^{\mathrm{N}}v_z^i \tag{S21}$$

Hence, the total muscle power simply equals the inertial power plus the aerodynamic power:

$$2{}^{\mathrm{N}}P^{\mathrm{pect}} + 2{}^{\mathrm{N}}P^{\mathrm{supra}} = {}^{\mathrm{N}}P^{\mathrm{inertia}} + {}^{\mathrm{N}}P^{\mathrm{aero}}, \tag{S22}$$

which could be computed using the aerodynamic forces and kinematics that we measured. During the mid-downstroke, the pectoralis primarily produces the power, so we eliminated the supracoracoideus term.

In order to convert the muscle power into force and stress, we incorporated the strain measured in vivo and other parameters measured during dissection. The power produced by the pectoralis equals the total force generated by the pectoralis measured at the attachment point, $F_{\mathrm{pect}}$, times the rate of change of the length of the entire pectoralis, $\dot{L}_{\mathrm{pect}}$:

$$^{\mathrm{N}}P^{\mathrm{pect}} = F_{\mathrm{pect}}\dot{L}_{\mathrm{pect}} \tag{S23}$$

Since the pectoralis is not uniform along its length, we computed the rate of change of its length using measurements taken from the in vivo experiment and during the dissection:

$$\dot{L}_{\mathrm{pect}} = cos\left(\bar{\alpha}_{\mathrm{pect,f}}\right)\dot{\gamma}_{\mathrm{pect,f}}\bar{L}_{\mathrm{pect,f}}, \tag{S24}$$

where $\bar{\alpha}_{\mathrm{pect,f}}$ is the average measured muscle fiber angle (dissection), $\bar{L}_{\mathrm{pect,f}}$ is the average measured muscle fiber length (dissection), and $\dot{\gamma}_{\mathrm{pect,f}}$ is the measured strain rate of the muscle fibers assuming uniform strain throughout the muscle (in vivo). To convert the muscle force to the average muscle stress, $\sigma_{\mathrm{pect}}$, we calculated the Physiological Cross-Sectional Area (PCSA) (**Alexander, 1983**), which is the area of the cross section of the pectoralis perpendicular to its fibers:

$$\sigma_{\mathrm{pect}} = \frac{F_{\mathrm{pect}}}{\mathrm{PCSA}_{\mathrm{pect}}}, \tag{S25}$$

$$\mathrm{PCSA}_{\mathrm{pect}} = \frac{m_{\mathrm{pect}}}{\rho_{\mathrm{pect}}\left(\bar{L}_{\mathrm{pect,f}}\right)} \tag{S26}$$

For muscle density, we use $\rho_{\mathrm{pect}} = 1060$ kg·m$^{-3}$ (**Tobalske and Biewener, 2008**), and $m_{\mathrm{pect}}$ is the average mass of the left and right pectorales of each dove measured during dissection.

## Section A8: 3D angular momentum balance of the wing

To compute the time-resolved 3D moment vector (torque) that the dove's muscles must have produced, we balanced the muscle moment with the effects of the aerodynamic forces and inertia. We used this result, together with the pectoralis force magnitude, computed by balancing 1D power, to compute the pull angle of the pectoralis on the humerus during downstroke.

Like the power balance derivation, we needed to decide what the most effective scope to analyze angular momentum is: a single wing of the dove, or the entire dove. We decided to analyze the angular momentum of a single wing of the dove because fortunately, whereas the reaction force at the shoulder joint is problematic for the power balance, it cancels out for the angular momentum balance. The base form of the 3D angular momentum balance equation is:

$$\vec{\mathbf{M}}^{\mathrm{W/O}} = \frac{{}^{\mathrm{N}}d{}^{\mathrm{N}}\vec{\mathbf{H}}^{\mathrm{W/O}}}{dt} + {}^{\mathrm{N}}\vec{\mathbf{v}}^{\mathrm{O}} \times {}^{\mathrm{N}}\vec{\mathbf{L}}^{\mathrm{W}}, \tag{S27}$$

where $\vec{\mathbf{M}}^{\mathrm{W/O}}$ is the sum of the moments on the wing (W: a system of particles and rigid bodies) about the shoulder joint (point O: fixed on the wing), ${}^{\mathrm{N}}\vec{\mathbf{H}}^{\mathrm{W/O}}$ is the angular momentum of the wing about the shoulder joint in the Newtonian reference frame, $\frac{{}^{\mathrm{N}}d}{dt}$ is the time derivative in the Newtonian reference frame, ${}^{\mathrm{N}}\vec{\mathbf{v}}^{\mathrm{O}}$ is the velocity of the shoulder joint in the Newtonian reference frame, and ${}^{\mathrm{N}}\vec{\mathbf{L}}^{\mathrm{W}}$ is the linear momentum of the wing in the Newtonian reference frame. Reaction forces at the shoulder joint do not create any moment about that shoulder joint, so we canceled that term. We eliminated any reaction torques by modelling the shoulder joint as a freely-rotating ball and socket joint with negligible friction. We also assumed negligible internal wing forces, such as muscles which

bend the wrist or elbow. With the scope of the angular momentum balance decided, we expanded each side of *Equation S27* to incorporate the effects of the muscles, aerodynamics, and inertia.

The terms on the right side of *Equation S27* all relate to the distributed mass of the wing. The linear momentum of the wing in the Newtonian reference frame is:

$$^{N}\vec{\mathbf{L}}^{W} = \sum_{i=1}^{I_{W}} m_{i} {}^{N}\vec{\mathbf{v}}^{i}, \tag{S28}$$

where $m_{i}$ is the $i^{th}$ of $I_{W}$ bodies (or point masses) on the wing, and $^{N}\vec{\mathbf{v}}^{i}$ is the velocity vector of the $i^{th}$ body in the Newtonian reference frame. The angular momentum of the wing about the shoulder joint in the Newtonian reference frame is:

$$^{N}\vec{\mathbf{H}}^{W/O} = \sum_{i=1}^{I_{W}} \left( \vec{\mathbf{I}}^{W_{i}} \cdot {}^{N}\vec{\boldsymbol{\omega}}^{W_{i}} + \vec{\mathbf{r}}^{i/O} \times m_{i} {}^{N}\vec{\mathbf{v}}^{i} \right), \tag{S29}$$

where $^{N}\vec{\boldsymbol{\omega}}^{W_{i}}$ is the angular velocity of the reference frame of the $i^{th}$ body in system W (the wing) relative to the Newtonian reference frame, $\vec{\mathbf{I}}^{W_{i}}$ is the moment of inertia of the $i^{th}$ body in system W relative to the center of mass of that body, and $\vec{\mathbf{r}}^{i/O}$ is the position vector from the shoulder joint to the $i^{th}$ body in system W. Because we modeled the dove's wings using point masses, we simplified to:

$$^{N}\vec{\mathbf{H}}^{W/O} = \sum_{i=1}^{I_{w}} \vec{\mathbf{r}}^{i/O} \times m_{i} {}^{N}\vec{\mathbf{v}}^{i} \tag{S30}$$

Next, we expanded the left-hand side of *Equation S27* which is the sum of the moments on the wing. This is made up of aerodynamic, gravitational, and muscle torques:

$$\vec{\mathbf{M}}^{W/O} = \vec{\mathbf{M}}_{pect}^{W/O} + \vec{\mathbf{M}}_{supra}^{W/O} - \vec{\mathbf{M}}_{aero}^{W/O} + \vec{\mathbf{M}}_{g}^{W/O}, \tag{S31}$$

where $\vec{\mathbf{M}}_{aero}^{W/O}$ is the aerodynamic moment, $\vec{\mathbf{M}}_{g}^{W/O}$ is the gravitational moment, $\vec{\mathbf{M}}_{pect}^{W/O}$ is the pectoralis moment, and $\vec{\mathbf{M}}_{supra}^{W/O}$ is the supracoracoideus moment. Each of these terms equals the cross product of the vector from the shoulder joint to the location where the force acts crossed with the force vector. Hence, the gravitational moment term can be written as:

$$\vec{\mathbf{M}}_{g}^{W/O} = \sum_{i=1}^{I_{w}} \vec{\mathbf{r}}^{i/O} \times (m_{i}\vec{\mathbf{g}}). \tag{S32}$$

This concludes the expansion of all terms related to mass in the angular momentum balance, and to summarize them, we defined a parameter called the inertial moment:

$$\vec{\mathbf{M}}_{inertia}^{W/O} = \frac{^{N}d^{N}\vec{\mathbf{H}}^{W/O}}{dt} + {}^{N}\vec{\mathbf{v}}^{O} \times {}^{N}\vec{\mathbf{L}}^{W} - \vec{\mathbf{M}}_{g}^{W/O}, \tag{S33}$$

$$\vec{\mathbf{M}}_{inertia}^{W/O} = \frac{^{N}d}{dt} \left( \sum_{i=1}^{I_{w}} \vec{\mathbf{r}}^{i/O} \times m_{i} {}^{N}\vec{\mathbf{v}}^{O} \right) + {}^{N}\vec{\mathbf{v}}^{O} \times \sum_{i=1}^{I_{w}} m_{i} {}^{N}\vec{\mathbf{v}}^{i} - \sum_{i=1}^{I_{w}} \vec{\mathbf{r}}^{i/O} \times (m_{i}\vec{\mathbf{g}}). \tag{S34}$$

By grouping terms in this manner, the total muscle moment equals the inertial moment plus the aerodynamic moment:

$$\vec{\mathbf{M}}_{pect}^{W/O} + \vec{\mathbf{M}}_{supra}^{W/O} = \vec{\mathbf{M}}_{inertia}^{W/O} + \vec{\mathbf{M}}_{aero}^{W/O}, \tag{S35}$$

similar to *Equation S22* for the power balance.

The aerodynamic moment was challenging to compute because aerodynamic forces acted continuously along the wing. In order to solve for the aerodynamic moment simply in terms of the aerodynamic force on the wing and known parameters, we modeled the wing using blade elements:

$$\vec{\mathbf{M}}_{\text{aero}}^{\text{W/O}} = -\sum_{j=1}^{J} \vec{\mathbf{r}}^{j/O} \times \vec{\mathbf{F}}_{\text{aero},j},$$

(S36)

where $\vec{\mathbf{r}}^{j/O}$ is the position vector from the shoulder joint to the quarter-chord location (a quarter of the way from the leading edge to the trailing edge) on the $j^{\text{th}}$ of J blade elements, and $\vec{\mathbf{F}}_{\text{aero},j}$ is the aerodynamic force on the $j^{\text{th}}$ blade element. We expanded $\vec{\mathbf{F}}_{\text{aero},j}$ to:

$$\vec{\mathbf{F}}_{\text{aero},j} = -0.5\rho S_j \left( C_{L,j}\hat{\mathbf{L}}_j + C_{D,j}\hat{\mathbf{D}}_j \right) \left| ^N\vec{\mathbf{v}}^j \right|^2,$$

(S37)

where the $j^{\text{th}}$ blade element has surface area, $S_j$, lift coefficient, $C_{L,j}$, lift direction, $\hat{\mathbf{L}}_j$, drag coefficient, $C_{D,j}$, drag direction, $\hat{\mathbf{D}}_j$, and velocity $^N\vec{\mathbf{v}}^j$. Assuming that the lift and drag coefficients and directions are spanwise invariant ($C_{L,j} = C_L$, $\hat{\mathbf{L}}_j = \hat{\mathbf{L}}$, $C_{D,j} = C_D$, and $\hat{\mathbf{D}}_j = \hat{\mathbf{D}}$ for all j), this reduces to:

$$\vec{\mathbf{M}}_{\text{aero}}^{W/O} = \sum_{j=1}^{J} \vec{\mathbf{r}}^{j/O} \times \left( 0.5\rho S_j \left( C_L\hat{\mathbf{L}} + C_D\vec{\mathbf{D}} \right) \left| ^N\vec{\mathbf{v}}^j \right|^2 \right),$$

(S38)

and further simplifies to:

$$\vec{\mathbf{M}}_{\text{aero}}^{W/O} = 0.5\rho \left[ \sum_{j=1}^{J} S_j \left| ^N\vec{\mathbf{v}}^j \right|^2 \vec{\mathbf{r}}^{j/O} \right] \times \left( C_L\hat{\mathbf{L}} + C_D\hat{\mathbf{D}} \right).$$

(S39)

In order to incorporate the total measured aerodynamic force on the wing, $\vec{\mathbf{F}}_{\text{aero}}$ into the equation for the aerodynamic moment, we simplified the aerodynamic moment to:

$$\vec{\mathbf{M}}_{\text{aero}}^{\text{W/O}} = -\vec{\mathbf{R}}^{\text{AM/O}} \times \vec{\mathbf{F}}_{\text{aero}} + \vec{\mathbf{T}}_O,$$

(S40)

where $\vec{\mathbf{R}}^{\text{AM/O}}$ is the position vector from the shoulder joint to the aerodynamic moment center, and $\vec{\mathbf{T}}_O$ accounts for extra torque due to paired, non-aligned forces. We assumed $\left| \vec{\mathbf{T}}_O \right| = 0$ based on the assumption that all of the lift and drag forces point in the same direction, and primarily act at locations along the wing that form a straight line starting at the shoulder joint. Using blade elements, we expanded $\vec{\mathbf{F}}_{\text{aero}}$, giving:

$$\vec{\mathbf{M}}_{\text{aero}}^{\text{W/O}} = \vec{\mathbf{R}}^{\text{AM/O}} \times \sum_{j=1}^{J} 0.5\rho S_j \left( C_{L,j}\hat{\mathbf{L}}_j + C_{D,j}\hat{\mathbf{D}}_j \right) \left| ^N\vec{\mathbf{v}}^j \right|^2,$$

(S41)

and then simplified by assuming spanwise invariance in lift and drag coefficients and direction:

$$\vec{\mathbf{M}}_{\text{aero}}^{\text{W/O}} = 0.5\rho \left( \sum_{j=1}^{J} S_j \left| ^N\vec{\mathbf{v}}^j \right|^2 \right) \vec{\mathbf{R}}^{\text{AM/O}} \times \left( C_{LL}\hat{\mathbf{L}} + C_D\hat{\mathbf{D}} \right).$$

(S42)

By comparing *Equations S39 and S42*, we solved for $\vec{\mathbf{R}}^{\text{AM/O}}$:

$$0.5\rho \left[ \sum_{j=1}^{J} S_j \left| ^N\vec{\mathbf{v}}^j \right|^2 \vec{\mathbf{r}}^{j/O} \right] \times \left( C_L\hat{\mathbf{L}} + C_D\hat{\mathbf{D}} \right) = 0.5\rho \left[ \sum_{j=1}^{J} S_j \left| ^N\vec{\mathbf{v}}^j \right|^2 \right] \vec{\mathbf{R}}^{\text{AM/O}} \times \left( C_L\hat{\mathbf{L}} + C_D\hat{\mathbf{D}} \right),$$

(S43)

$$\vec{\mathbf{R}}^{\text{AM/O}} = \frac{\sum_{j=1}^{J} S_j \left| ^N\vec{\mathbf{v}}^j \right|^2 \vec{\mathbf{r}}^{j/O}}{\sum_{j=1}^{J} S_j \left| ^N\vec{\mathbf{v}}^j \right|^2}.$$

(S44)

The full expansion of the aerodynamic moment is:

$$\vec{\mathbf{M}}_{\mathrm{aero}}^{\mathrm{W/O}} = -\left(\frac{\sum\limits_{j=1}^{J} S_j \left|^{\mathrm{N}}\vec{\mathbf{v}}^j\right|^2 \vec{\mathbf{r}}^{j/O}}{\sum\limits_{j=1}^{J} S_j \left|^{\mathrm{N}}\vec{\mathbf{v}}^j\right|^2}\right) \times \vec{\mathbf{F}}_{\mathrm{aero}}, \tag{S45}$$

which closely resembles *Equation S17* for aerodynamic power. By incorporating this equation for the aerodynamic moment, *Equation S35* can be used to solve for the moment that the flight muscles together must generate.

In order to compute the angle that the pectoralis pulls on the humerus, we used the computed pectoralis force magnitude from the 1D power balance and the muscle moment from the 3D angular momentum balance. We started by expanding the pectoralis moment:

$$\vec{\mathbf{M}}_{\mathrm{pect}}^{\mathrm{W/O}} = \vec{\mathbf{r}}^{\mathrm{P/O}} \times \vec{\mathbf{F}}_{\mathrm{pect}}, \tag{S46}$$

where $\vec{\mathbf{r}}^{\mathrm{P/O}}$ is the vector from the shoulder joint to the deltopectoral crest where the pectoralis pulls on the humerus, and where $\vec{\mathbf{F}}_{\mathrm{pect}}$ is the total pectoralis force. While the directions of $\vec{\mathbf{r}}^{\mathrm{P/O}}$ and $\vec{\mathbf{F}}_{\mathrm{pect}}$ were unknown and changed continuously through the stroke, we knew the magnitudes of both: $\left|\vec{\mathbf{r}}^{\mathrm{P/O}}\right|$ from dissection and $\left|\vec{\mathbf{F}}_{\mathrm{pect}}\right|$ from the 1D power balance. Hence, we used the following equation, derived from *Equation S46*, to solve for the effective pull ratio, $\sin\theta_p$, and the pectoralis pull angle on the humerus, $\theta_p$:

$$\left|\vec{\mathbf{M}}_{\mathrm{pect}}^{\mathrm{W/O}}\right| = \sin\theta_p \left|\vec{\mathbf{r}}^{\mathrm{P/O}}\right| \left|\vec{\mathbf{F}}_{\mathrm{pect}}\right|, \tag{S47}$$

$$\theta_p = \sin^{-1}\left(\frac{\left|\vec{\mathbf{M}}_{\mathrm{pect}}^{\mathrm{W/O}}\right|}{\left|\vec{\mathbf{r}}^{\mathrm{P/O}}\right| \left|\vec{\mathbf{F}}_{\mathrm{pect}}\right|}\right). \tag{S48}$$

The next section details how we used this result to compute the full 3D pectoralis force vector.

## Section A9: Determining the pectoralis pull direction

To determine the direction in which the pectoralis pulls on the humerus, we overlaid the skeleton of a dove on the measured 3D surface (*Figure 6*, *Figure 6—figure supplement 1*). Using Computed Tomography (CT), we scanned a dove skeleton of similar mass (170.2 g) to the four doves in this study, and then scaled the dimensions for each of the four doves to match the size of the dove we CT scanned. The scaling factor was the one third power of the ratio of each of the four dove's masses to the CT-scanned dove's mass. We assumed that the bones in the body of the dove (ribs, sternum, furcula, coracoid, and scapula) do not move relative to each other during flight. In reality, they do move, but not substantially compared to the movement of the wings (*Baier et al., 2013*; *Heers et al., 2016*). We also assumed that the bones in the wing consist of three rigid bone segments connected to each other: (1) the brachium (humerus), (2) the antebrachium (radius and ulna), and (3) the manus (radiale +ulnare + carpometacarpus +phalanges). The positions and orientations of the reference frames (joint axes) associated with each bone segment were determined based on the inertial axes of the bones and the joint anatomy (*Baier et al., 2013*; *Heers et al., 2016*). We assumed that all wing joints were free to rotate with 3 degrees of freedom and no translational degrees of freedom (measurements show that translation of the joints relative to the reference frame is minimal for modeling purposes) (*Baier et al., 2013*; *Heers et al., 2016*). The positioning of each joint was determined based on the centers of rotation (*Stowers, 2017*) when previously-measured kinematics in chukars (*Alectoris chukar*; kinematics measured using X-Ray Reconstruction of Moving Morphology, i.e., XROMM) (*Heers et al., 2016*) were applied to the dove model. Muscle paths were modeled based on dissection of the CT-scanned dove. Pins were pushed into the body through the pectoralis and supracoracoideus in multiple locations along the central tendons, and pictures were taken in situ for multiple wing positions. Then the pectoralis and supracoracoideus were removed from the body, and the positions of the pins were re-photographed to visualize the paths of the tendons with respect to the skeleton. These paths were used to construct a simplified musculoskeletal model in SIMM (Software for Interactive Musculoskeletal Modeling), using wrapping surfaces to prevent the muscles from passing through bone.

We overlaid the musculoskeletal model, made up of four skeletal sections (one for the body and three for the left wing) onto the 3D surface of the dove during each flight in four stages.

1. First, we determined the position and orientation of the skeleton relative to the 3D surface of the body. We matched the orientation of the vertebral column with the pitch and yaw angles of 3D surface fit of the body (assuming zero roll), and we matched the plane of symmetry of the skeletal and 3D surface bodies. We positioned the skeletal shoulder joint as close as possible to the average position (relative to the body of the dove) of the manually tracked shoulder joint, subject to the previous constraints.

2. Second, we rotated the humerus about the shoulder joint (rigidly attached to the dove's body) and the radius/ulna about the elbow joint (rigidly attached to the end of the humerus) until the end of the radius/ulna (the wrist joint) best matched the manually tracked wrist of the flying doves. Since rotation about the long axis of the humerus and the long axis of the radius/ulna do not impact the final position of the wrist from a purely geometric perspective (in reality, long axis rotation of the humerus partially determines how much abduction or adduction occurs at the elbow, which in turn determines the position of the wrist), this left two rotational axes of consequence for each bone segment: hence four rotational parameters to fit. Constraining the wrist joint to match between the skeletal and 3D surface models generates three constraints. For the fourth constraint, we rotated the humerus and radius/ulna so the deltopectoral crest and dorsal surfaces of the radius and ulna were approximately in the plane of the 3D surface of the wing. We then solved for these four rotational angles using a nonlinear least-squares regression, and smoothed the result (**Eilers, 2003**), to find the orientations of the humerus and radius/ulna.

3. Third, we oriented the manus such that the most distal phalanx pointed toward the ninth primary wingtip (which was tracked).

4. Fourth, we allowed the humerus to rotate in the long axis so that the moment generated by the pectoralis best matched the required muscle moment, $\vec{\mathbf{M}}_{\mathrm{pect}}^{\mathrm{W/O}}$ when the pectoralis was shortening.

Using the overlaid skeleton, the pectoralis force magnitude, the pectoralis moment, and the pectoralis pull angle, we computed the time-resolved 3D force vector of the pectoralis. Examining **Equation S46**, the overlaid skeleton provides the missing information necessary to determine the vector from the shoulder joint to the deltopectoral crest where the pectoralis pulls on the humerus, $\vec{\mathbf{r}}^{\mathrm{P/O}}$. With this information, we solved for the direction of the pectoralis force, $\vec{\mathbf{F}}_{\mathrm{pect}}/\left|\vec{\mathbf{F}}_{\mathrm{pect}}\right|$:

$$\frac{\vec{\mathbf{F}}_{\mathrm{pect}}}{\left|\vec{\mathbf{F}}_{\mathrm{pect}}\right|} = -R\left(\frac{\vec{\mathbf{M}}_{\mathrm{pect}}^{\mathrm{W/O}}}{\left|\vec{\mathbf{M}}_{\mathrm{pect}}^{\mathrm{W/O}}\right|}, -\theta_{\mathrm{p}}\right) \cdot \frac{\vec{\mathbf{r}}^{\mathrm{P/O}}}{\left|\vec{\mathbf{r}}^{\mathrm{P/O}}\right|}, \tag{S49}$$

where $R\left(\mathbf{a}, \theta\right)$ is a function which outputs a rotation matrix corresponding to a rotation about axis $\mathbf{a}$ by an angle of $\theta$. In other words, the direction of $\vec{\mathbf{F}}_{\mathrm{pect}}$ is found by rotating the negative direction of $\vec{\mathbf{r}}^{\mathrm{P/O}}$ about the direction of $\vec{\mathbf{M}}_{\mathrm{pect}}^{\mathrm{W/O}}$ by an angle of $-\theta_{\mathrm{p}}$. Note that in order to compute the direction of $\vec{\mathbf{r}}^{\mathrm{P/O}}$, only the orientation of the humerus is needed. The other skeletal segments (radius/ulna and manus) are computed for visualization purposes in the figures.

## Section A10: Rules for modeling muscle power overlap

To address the modeling challenge of disentangling pectoralis and supracoracoideus muscle power during stroke-reversal, we considered the case where there is no elastic energy storage, and the muscles only generate or absorb power. Based on the sonomicrometry measurements of the pectoralis and total computed muscle power, each time-step during the stroke fell into one of four categories: (1) Positive total muscle power while the pectoralis was shortening. This was the case for much of the downstroke and indicated that the pectoralis was generating power. However, near stroke reversal, low pectoralis muscle velocity prevented high levels of power generation (see **Equation S23**). Hence, we restricted the maximum force production of pectoralis in stroke-reversal regions based on the maximum force produced by the pectoralis during the mid-downstroke. Any leftover total power is categorized as supracoracoideus power generation. (2) Negative total muscle power while the pectoralis was shortening. This was the case at the end of the upstroke and indicated that the supracoracoideus was acting as a brake by absorbing power. (3) Positive total muscle power while the pectoralis was lengthening. This was the case for much of the upstroke and indicated that the supracoracoideus was generating power. (4) Negative total muscle power while

the pectoralis was lengthening. This was the case during the latter half of the upstroke when the wing was decelerating and indicated that the pectoralis was acting as a brake and absorbing power. However, similar to case 1, if pectoralis muscle velocity was too low to absorb the entirety of the total muscle power, we attributed the remainder to supracoracoideus power absorption.

## Section A11: Modeling energy storage in the supracoracoideus tendon

To model energy storage in the supracoracoideus tendon, we considered how different amounts and timing of elastic energy storage in the tendon would impact the power distribution between the pectoralis and supracoracoideus. The two tuning parameters we varied are: (1) the proportion of supracoracoideus power which was elastically stored and released, and (2) the time period over which energy was stored in the supracoracoideus tendon. When we increased the first tuning parameter, we proportionally shifted the distribution of supracoracoideus power from generated to released (upstroke), and from absorbed to stored (downstroke). The power generated by the pectoralis was correspondingly increased to ensure sufficient energy for the supracoracoideus tendon to store during the downstroke. To account for hysteresis effects (*Tobalske and Biewener, 2008*) in the supracoracoideus tendon, the integrated extra power produced by the pectoralis must exceed the energy released by the supracoracoideus tendon by 7%. In order to estimate the temporal distribution of energy storage in the supracoracoideus, we assumed that the supracoracoideus lengthens according to measurements of ascending flight in pigeons (Figure 2 of *Tobalske and Biewener, 2008*). Specifically, we matched the timing of the pectoralis strain between our measurements and their measurements, in order to match the supracoracoideus strain and find when the supracoracoideus finishes lengthening. This timing, combined with our timing tuning parameter, defined the region of the stroke over which the supracoracoideus stores energy. Next, we assumed that the shape of the force-length curve of the supracoracoideus tendon matched the curve given in *Millard et al., 2013*. With this timing, strain, and force shape information, we calculated the shape of the power curve for the supracoracoideus tendon as a function of stroke percentage. We then scaled the magnitude of the power curve according to the amount of energy the supracoracoideus needed to store. By varying these two tuning parameters, we examined the effects of elastic energy storage in the supracoracoideus tendon on muscle performance.

## Section A12: Scaling analysis across extant birds

We scaled our aerodynamic, inertial, and muscle measurements for doves across multiple extant birds, to estimate comparative patterns of power output during slow flight.

For each extant bird analyzed, we scaled our results for doves by modifying all relevant variables (see *Supplementary file 2* and *Supplementary file 3* for details), and plugging them directly into the relevant equations. We scaled the following variables based on extant data of 27 birds measured in literature (*Berg and Rayner, 1995*): body mass, $m_{\text{body}}$, wing mass (single wing), $m_{\text{wing}}$, wing moment of inertia (single wing), $I_{\text{wing}}$, wingspan (distance from shoulder joint to wingtip; single wing), $|\vec{\mathbf{r}}_{\text{span}}|$, distance from the shoulder joint to the center of gravity of the wing (single wing), $|\vec{\mathbf{r}}_{\text{wing,cg}}|$, wing area (single wing), $S_{\text{wing}}$, and wingbeat frequency, $f$. For four additional birds where data was missing, we used data from other sources (*Jackson and Dial, 2011a*; *Greenewalt, 1962*), scaled isometrically to match the body mass. For pectoralis mass (single pectoralis), $m_{\text{pect}}$, we isometrically scaled data in literature to match the body mass (*Tobalske and Biewener, 2008*; *Tobalske et al., 2005*; *Greenewalt, 1962*). Finally, we assumed the following relationships for the following variables: time step for integration, $\Delta t$:

$$\Delta t \propto f^{-1},\tag{S50}$$

aerodynamic force, $\vec{\mathbf{F}}_{\text{aero}}$:

$$\vec{\mathbf{F}}_{\text{aero}} \propto m_{\text{body}},\tag{S51}$$

wing velocity (distributed), $^{\text{N}}\vec{\mathbf{v}}_{\text{wing}}^{\text{i}}$:

$$^{\text{N}}\vec{\mathbf{v}}_{\text{wing}}^{\text{i}} \propto f\,|\vec{\mathbf{r}}_{\text{span}}|,\tag{S52}$$

and body velocity, $^{\text{N}}\vec{\mathbf{v}}_{\text{body}}$ equivalent to the doves. We calculated the point mass distribution in the wing, $m_{\text{wing,i}}$, using a least-squares fit to match wing mass, inertia, and center of gravity

data. Additionally, to compare pectoralis electrical activation timing to our scaling parameters in *Figure 7D*, we used the same procedure for 17 birds (*Supplementary file 2b* and *Supplementary file 3b-c*) (*Tobalske and Biewener, 2008*; *Biewener et al., 1992*; *Jackson and Dial, 2011a*; *Dial and Biewener, 1993*; *Hedrick et al., 2003*; *Tobalske et al., 2005*; *Berg and Rayner, 1995*; *Tobalske et al., 2017*; *Altshuler et al., 2010*; *Tobalske et al., 2010*; *Hedrick et al., 2004*; *Ingersoll and Lentink, 2018*; *Jackson et al., 2011b*; *Donovan et al., 2013*; *Greenewalt, 1962*; *Dial et al., 1997*).

To interpret the scaling relationships, we traced the effects of scaling parameters in the aerodynamic power and moment equations. For aerodynamic power, we started with *Equations S1 and S17* combined for some bird:

$$^{\mathrm{N}}P^{\mathrm{aero}} = 2\vec{\mathbf{F}}_{\mathrm{aero}} \cdot \left( \frac{\sum_{j=1}^{J} S_j \left| ^{\mathrm{N}}\vec{\mathbf{v}}^j \right|^2 {}^{\mathrm{N}}\vec{\mathbf{v}}^j}{\sum_{j=1}^{J} S_j \left| ^{\mathrm{N}}\vec{\mathbf{v}}^j \right|^2} \right) \tag{S53}$$

Next, for each variable, we replaced it with an equivalent expression by utilizing the scaling relationships:

$$^{\mathrm{N}}P^{\mathrm{aero}} = 2\tilde{\vec{\mathbf{F}}}_{\mathrm{aero}} \frac{m_{\mathrm{body}}}{\tilde{m}_{\mathrm{body}}} \cdot \left( \frac{\sum_{j=1}^{J} \tilde{S}_j \frac{S_{\mathrm{wing}}}{\tilde{S}_{\mathrm{wing}}} \left| ^{\mathrm{N}}\tilde{\vec{\mathbf{v}}}^j \frac{f \left| \vec{\mathbf{r}}_{\mathrm{span}} \right|}{\tilde{f} \left| \tilde{\vec{\mathbf{r}}}_{\mathrm{span}} \right|} \right|^2 {}^{\mathrm{N}}\tilde{\vec{\mathbf{v}}}^j \frac{f \left| \vec{\mathbf{r}}_{\mathrm{span}} \right|}{\tilde{f} \left| \tilde{\vec{\mathbf{r}}}_{\mathrm{span}} \right|}}{\sum_{j=1}^{j} \tilde{S}_j \frac{S_{\mathrm{wing}}}{\tilde{S}_{\mathrm{wing}}} \left| ^{\mathrm{N}}\tilde{\vec{\mathbf{v}}}^j \frac{f \left| \mathbf{r}_{\mathrm{span}} \right|}{\tilde{f} \left| \tilde{\vec{\mathbf{r}}}_{\mathrm{span}} \right|} \right|^2} \right), \tag{S54}$$

where variables with a tilde represent a second bird. When we reduced this expression, we recovered the equation for aerodynamic power for the second bird:

$$^{\mathrm{N}}P^{\mathrm{aero}} = \left( \frac{m_{\mathrm{body}} \, f \left| \vec{\mathbf{r}}_{\mathrm{span}} \right|}{\tilde{m}_{\mathrm{body}} \, \tilde{f} \left| \tilde{\vec{\mathbf{r}}}_{\mathrm{span}} \right|} \right) 2\vec{\mathbf{F}}_{\mathrm{aero}} \cdot \left( \frac{\sum_{j=1}^{J} \tilde{S}_j \left| ^{\mathrm{N}}\tilde{\vec{\mathbf{V}}}^j \right|^2 {}^{\mathrm{N}}\tilde{\vec{\mathbf{V}}}^j}{\sum_{j=1}^{J} \tilde{S}_j \left| ^{\mathrm{N}}\tilde{\vec{\mathbf{V}}}^j \right|^2} \right), \tag{S55}$$

which led us to the scaling relationship for aerodynamic power:

$$\frac{^{\mathrm{N}}P^{\mathrm{aero}}}{^{\mathrm{N}}\tilde{P}^{\mathrm{aero}}} = \frac{m_{\mathrm{body}} \, f \left| \vec{\mathbf{r}}_{\mathrm{span}} \right|}{\tilde{m}_{\mathrm{body}} \, \tilde{f} \left| \tilde{\vec{\mathbf{r}}}_{\mathrm{span}} \right|}, \tag{S56}$$

$$^{\mathrm{N}}P^{\mathrm{aero}} \propto m_{\mathrm{body}} f \left| \vec{\mathbf{r}}_{\mathrm{span}} \right| \tag{S57}$$

We then rearranged this equation to form interpretable parameters:

$$\frac{^{\mathrm{N}}P^{\mathrm{aero}}}{m_{\mathrm{body}}} \propto f \left| \vec{\mathbf{r}}_{\mathrm{span}} \right|, \tag{S58}$$

where $f \left| \vec{\mathbf{r}}_{\mathrm{span}} \right|$ is proportional to wingtip speed. Using the same technique, we find the scaling relationship for the aerodynamic moment:

$$\frac{\vec{\mathbf{M}}_{\mathrm{aero}}^{\mathrm{W/O}}}{m_{\mathrm{body}}} \propto \left| \vec{\mathbf{r}}_{\mathrm{span}} \right| \tag{S59}$$

Interpreting the effects of the scaling relationships on the inertial power and moments was more challenging. For inertial power, we started with *Equation S21* for some bird:

$$^{\text{N}}P^{\text{inertia}} = \frac{1}{2}\frac{d}{dt}m_{\text{body}}\,{}^{\text{N}}\vec{\mathbf{v}}_{\text{body}} \cdot {}^{\text{N}}\vec{\mathbf{v}}_{\text{body}} + \frac{1}{2}\frac{d}{dt}\left(\sum_{i=1}^{I_{\text{w}}} m_{\text{wing,i}}\,{}^{\text{N}}\vec{\mathbf{v}}_{\text{wing}}^{\text{i}} \cdot {}^{\text{N}}\vec{\mathbf{v}}_{\text{wing}}^{\text{i}}\right) + \sum_{i=1}^{I_{\text{W,B}}} m_{\text{i}}g\,{}^{\text{N}}v_{z}^{\text{i}},$$ (S60)

where the point masses representing the body and wings were expanded from *Equation S21*. The gravity power was negligible so we ignored it, and the scaling relationship for the inertial power due to the movement of the body is simple:

$$^{\text{N}}P_{\text{body}}^{\text{inertia}} \propto m_{\text{body}},$$ (S61)

because the body velocity was unchanged in our scaling analysis. However, the inertial term for the wing is more complex because of the time derivative and because the point masses do not scale proportionately. To incorporate the moment of inertia of the wing, which does scale proportionately, we expanded the velocity terms. We started by assuming that the stroke angle, $\phi$, could be written as:

$$\phi = -\Phi\cos\left(2\pi ft\right),$$ (S62)

where $\Phi$ is the stroke amplitude angle. We then differentiated the stroke angle so that the velocity of the point masses could be written as:

$$^{\text{N}}\vec{\mathbf{v}}_{\text{wing}}^{\text{i}} = \dot{\phi}\vec{\mathbf{r}}_{\text{wing}}^{\text{i/O}} = \Phi 2\pi f\sin\left(2\pi ft\right)\vec{\mathbf{r}}_{\text{wing}}^{\text{i/O}}$$ (S63)

When we plugged this into the power equation:

$$^{\text{N}}P_{\text{wing}}^{\text{inertia}} = \frac{1}{2}\frac{d}{dt}\left(\sum_{i=1}^{I_{\text{w}}} m_{\text{wing,i}}\left|\Phi 2\pi f\sin\left(2\pi ft\right)\vec{\mathbf{r}}_{\text{wing}}^{\text{i/O}}\right|^{2}\right),$$ (S64)

and simplified:

$$^{\text{N}}P_{\text{wing}}^{\text{inertia}} = \frac{1}{2}\left(\Phi 2\pi f\right)^{2}\frac{d}{dt}\left(\sin^{2}\left(2\pi ft\right)\sum_{i=1}^{I_{\text{w}}} m_{\text{wing,i}}\left|\vec{\mathbf{r}}_{\text{wing}}^{\text{i/O}}\right|^{2}\right),$$ (S65)

we found that the moment of inertia of the wing appeared:

$$^{\text{N}}P_{\text{wing}}^{\text{inertia}} = \frac{1}{2}\left(\Phi 2\pi f\right)^{2}\frac{d}{dt}\left(\sin^{2}\left(2\pi ft\right)I_{\text{wing}}\right)$$ (S66)

After differentiating the expression, we arrived at the following simplified equation:

$$^{\text{N}}P_{\text{wing}}^{\text{inertia}} = 8\Phi^{2}\pi^{3}f^{3}I_{\text{wing}}\sin\left(2\pi ft\right),$$ (S67)

from which we can easily see that:

$$^{\text{N}}P_{\text{wing}}^{\text{inertia}} \propto f^{3}I_{\text{wing}}$$ (S68)

Assuming that the inertial power from the wing dominated the inertial power from the body, we can summarize in the following interpretable format:

$$\frac{^{\text{N}}P^{\text{inertia}}}{m_{\text{body}}} \propto \frac{m_{\text{wing}}}{m_{\text{body}}}\left(\frac{r_{\text{gyr}}}{\left|\vec{\mathbf{r}}_{\text{span}}\right|}\right)^{2}\left(f\left|\vec{\mathbf{r}}_{\text{span}}\right|\right)^{2}f,$$ (S69)

where $r_{\text{gyr}}$ is the radius of gyration of the wing ($I_{\text{wing}} = m_{\text{wing}}r_{\text{gyr}}^{2}$). For the inertial moment, we conducted a similar analysis, assuming that the angular momentum term dominated, producing the following scaling relationship:

$$\frac{\vec{\mathbf{M}}_{\text{inertia}}^{\text{W/O}}}{m_{\text{body}}} \propto \frac{m_{\text{wing}}}{m_{\text{body}}}\left(\frac{r_{\text{gyr}}}{\left|\vec{\mathbf{r}}_{\text{span}}\right|}\right)^{2}\left(f\left|\vec{\mathbf{r}}_{\text{span}}\right|\right)^{2}$$ (S70)

For easier discussion of the different scaling factors which relate to inertial and aerodynamic power and moments, we defined new scaling parameters. We defined the aerodynamic power scaling parameter, $x_{\mathrm{p,aero}}$, which is proportional to the wingtip velocity as:

$$x_{\mathrm{p,aero}} = f\left|\vec{\mathbf{r}}_{\mathrm{span}}\right| \tag{S71}$$

We defined the aerodynamic moment scaling parameter, $x_{\mathrm{m,aero}}$, as the wingspan:

$$x_{\mathrm{m,aero}} = \left|\vec{\mathbf{r}}_{\mathrm{span}}\right| = \frac{x_{\mathrm{p,aero}}}{f} \tag{S72}$$

We defined the inertial power scaling parameter, $x_{\mathrm{p,iner}}$, as proportional to the product of the wing mass ratio, times the wing radius of gyration ratio squared, times the wingtip velocity squared, times the wingbeat frequency:

$$x_{\mathrm{p,iner}} = \frac{m_{\mathrm{wing}}}{m_{\mathrm{body}}} \left(\frac{r_{\mathrm{gyr}}}{\left|\vec{\mathbf{r}}_{\mathrm{span}}\right|}\right)^2 \left(f\left|\vec{\mathbf{r}}_{\mathrm{span}}\right|\right)^2 f \tag{S73}$$

Finally, we defined the inertial moment scaling parameter, $x_{\mathrm{m,iner}}$, as proportional to the product of the wing mass ratio, times the wing radius of gyration ratio squared, times the wingtip velocity squared:

$$x_{\mathrm{m,iner}} = \frac{m_{\mathrm{wing}}}{m_{\mathrm{body}}} \left(\frac{r_{\mathrm{gyr}}}{\left|\vec{\mathbf{r}}_{\mathrm{span}}\right|}\right)^2 \left(f\left|\vec{\mathbf{r}}_{\mathrm{span}}\right|\right)^2 = \frac{x_{\mathrm{p,iner}}}{f} \tag{S74}$$

Additionally, in **Figure 7B and D** we use the following expression to quantify the midway point of the pectoralis power exertion, $T_{\mathrm{P,mid}}^{\mathrm{pect}}$:

$$T_{\mathrm{P,mid}}^{\mathrm{pect}} = \frac{\int_{t_{\mathrm{Down-start}}}^{t_{\mathrm{Down-end}}} {}_{\mathrm{N}}P^{\mathrm{pect}} t\,dt}{\int_{t_{\mathrm{Down-start}}}^{t_{\mathrm{Down-end}}} {}_{\mathrm{N}}P^{\mathrm{pect}}\,dt} \times 100\%, \tag{S75}$$

and the midway point of the pectoralis force exertion, $T_{\mathrm{F,mid}}^{\mathrm{pect}}$:

$$T_{\mathrm{F,mid}}^{\mathrm{pect}} = \frac{\int_{t_{\mathrm{Down-start}}}^{t_{\mathrm{Down-end}}} {}_{\mathrm{N}}F^{\mathrm{pect}} t\,dt}{\int_{t_{\mathrm{Down-start}}}^{t_{\mathrm{Down-end}}} {}_{\mathrm{N}}F^{\mathrm{pect}}\,dt} \times 100\%, \tag{S76}$$

where $t_{\mathrm{Down-start}}$ and $t_{\mathrm{Down-end}}$ are the starting and ending times of the downstroke. To convert the ratio into a percentage, we multiply by 100%, since these points represent timing relative to stroke percentage.

