## [Editor Report · eLife assessment]

This **important** study combines experiments and mathematical modeling to enhance our understanding of the interplay between the two flight muscles in birds during slow flight. The evidence for the findings is **compelling**, derived from new methods for measuring wing shape and force production combined with previously validated methods in muscle physiology. This work will be of broad interest to comparative biomechanists.

---

## [Referee Report · Reviewer #1 (Public Review)]

The authors sought to resolve the coordinated functions of the two muscles that primarily power flight in birds (supracoracoideus and pectoralis), with particular focus on the pectoralis. Technology has limited the ability to resolve some details of pectoralis function, so the authors developed a model that can make accurate predictions about this muscle's function during flight. The authors first measured aerodynamic forces, wing shape changes, and pectoralis muscle activity in flying doves. They used cutting-edge techniques for the aerodynamic and wing shape measurements and they used well-established methods to measure activity and length of the pectoralis muscle. The authors then developed two mathematical models to estimate the instantaneous force vector produced by the pectoralis throughout the wing stroke. Finally, the authors applied their mathematical models to other-sized birds in order to compare muscle physiology across species.

The strength of the methods is that they smoothly incorporate techniques from many complementary fields to generate a comprehensive model of pectoralis muscle function during flight. The high-speed structured-light technique for quantifying surface area during flight is novel and cutting-edge, as is the aerodynamic force platform used. These methods push the boundaries of what has historically been used to quantify their respective aspects of bird flight and their use here is exciting. The methods used for measuring muscle activation and length are standard in the field. Together, these provide both a strong conceptual foundation for the model and highlight its novelty. This model allows for estimations of muscle function that are not feasible to measure in live birds during flight at present. The weakness of this approach is that it relies heavily on a series of assumptions. While the research presented in this paper makes use of powerful methods from multiple fields, those methods each have assumptions inherent to them that simplify the biological system of study. This reduction in the complexity of phenomena allows specific measurements to be made. In joining the techniques of multiple fields to study greater complexity of the phenomenon of interest, the assumptions are all incorporated also. Furthermore, assumptions are inherent to mathematical modelling of biological phenomena. That being said, the authors acknowledge and justify their assumptions at each step and their model seems to be quite good at predicting muscle function.

Indeed, the authors achieve their aims. They effectively integrate methods from multiple disciplines to explore the coordination and function of the pectoralis and supracoracoideus muscles during flight. The conclusions that the authors derive from their model address the intended research aim.

The authors demonstrate the value of such interdisciplinary research, especially in studying complex behaviors that are difficult or infeasible to measure in living animals. Additionally, this work provides predictions for muscle function that can be tested empirically. These methods are certainly valuable for understanding flight, but also have implications for biologists studying movement and muscle function more generally.

---

## [Referee Report · Reviewer #2 (Public Review)]

In this work, the authors investigated the pectoralis work loop and the function of the supracoracoideus muscle in the down stroke during slow flight in doves. The aim of this study was to determine how aerodynamic force is generated, using simultaneous high-speed measurements of the wings' kinematics, aerodynamics, and activation and strain of pectoralis muscles during slow flight. The measurements show a reduction in the angle of attack during mid-downstroke, which induces a peak power factor and facilitates the tensioning of the supracoracoideus tendon with pectoralis power, which then can be released in the up-stroke. By combining the data with a muscle mechanics model, the timely tuning of elastic storage in the supracoracoideus tendon was examined and showed an improvement of the pectoralis work loop shape factor. Finally, other bird species were integrated into the model for a comparative investigation.

The major strength of the methods is the simultaneous application of four high-speed techniques - to quantify kinematics, aerodynamics and muscle activation and strain - as well as the implementation of the time-resolved data into a muscle mechanics model. With a thorough analysis which supports the conclusions convincingly, the authors achieved their goal of reaching an improved understanding of the interplay of the pectoralis and supracoracoideus muscles during slow flight and the resulting energetic benefits.

---

## [Author Response]

The following is the authors’ response to the original reviews.

Response to Reviewers:

Thank you for taking the time to review our manuscript and provide us with helpful comments. Your comments enabled us to improve the clarity of the manuscript, in particular:

1. We improved the organization of the figures by associating each supplemental figure with a main-text figure using the eLife “figure supplements” format.

2. We reduced the length of figure captions where possible.

3. We improved organizational clarity by adding a brief organizational summary statement at the beginning of the results section which outlines the contents of the results subsections in the context of the introduction. We took particular care to use the same language, so the parallelism is clearer.

4. In addition, we made various modifications to the main text to improve clarity for the reader. For this we asked specific help of our biologist co-authors to indicate which aspects would benefit from further clarification to enable the broad biology readership of eLife to comprehend our research better.

**Reviewer #1 (Public Review):**
The authors sought to resolve the coordinated functions of the two muscles that primarily power flight in birds (supracoracoideus and pectoralis), with particular focus on the pectoralis. Technology has limited the ability to resolve some details of pectoralis function, so the authors developed a model that can make accurate predictions about this muscle's function during flight. The authors first measured aerodynamic forces, wing shape changes, and pectoralis muscle activity in flying doves. They used cutting-edge techniques for the aerodynamic and wing shape measurements and they used well-established methods to measure activity and length of the pectoralis muscle. The authors then developed two mathematical models to estimate the instantaneous force vector produced by the pectoralis throughout the wing stroke. Finally, the authors applied their mathematical models to other-sized birds in order to compare muscle physiology across species.The strength of the methods is that they smoothly incorporate techniques from many complementary fields to generate a comprehensive model of pectoralis muscle function during flight. The high-speed structured-light technique for quantifying surface area during flight is novel and cutting-edge, as is the aerodynamic force platform used. These methods push the boundaries of what has historically been used to quantify their respective aspects of bird flight and their use here is exciting. The methods used for measuring muscle activation and length are standard in the field. Together, these provide both a strong conceptual foundation for the model and highlight its novelty. This model allows for estimations of muscle function that are not feasible to measure in live birds during flight at present. The weakness of this approach is that it relies heavily on a series of assumptions. While the research presented in this paper makes use of powerful methods from multiple fields, those methods each have assumptions inherent to them that simplify the biological system of study. This reduction in the complexity of phenomena allows the specific measurements to be made. In joining the techniques of multiple fields to study the greater complexity of the phenomenon of interest, the assumptions are all incorporated also. Furthermore, assumptions are inherent to mathematical modeling of biological phenomena. That being said, the authors acknowledge and justify their assumptions at each step and their model seems to be quite good at predicting muscle function.Indeed, the authors achieve their aims. They effectively integrate methods from multiple disciplines to explore the coordination and function of the pectoralis and supracoracoideus muscles during flight. The conclusions that the authors derive from their model address the intended research aim.The authors demonstrate the value of such interdisciplinary research, especially in studying complex behaviors that are difficult or infeasible to measure in living animals. Additionally, this work provides predictions for muscle function that can be tested empirically. These methods are certainly valuable for understanding flight but also have implications for biologists studying movement and muscle function more generally.

Thank you for your thorough and positive review. We appreciate that you read our manuscript carefully and gave detailed feedback.

**Recommendations For The Authors:**
I thought that your manuscript was very interesting and your integration of techniques from multiple fields was effective. You address the weaknesses I highlighted in the public review well throughout the manuscript.

Thank you for your well-measured feedback on this weakness and how we addressed it.

I sometimes found that the manuscript was difficult to follow. With the interdisciplinary nature of your work, your manuscript has a lot of complexity. Your introduction is clear and I think that the last paragraph outlines your study very well. In the subsequent sections, the sub-headings are helpful, but I think your manuscript could be improved by indicating where those subsections fit into the phases you outline in your introduction (namely, muscle function, kinematics and aerodynamics, and mathematical modeling).

Complied: throughout the manuscript we made modifications to improve the clarity. We also added a brief organizational summary statement at the beginning of the results section which outlines the contents of the results section in the context of the language introduced in the introduction. Finally, we reorganized the supplemental figures into eLife’s favored format of “figure supplements”, so that each extra figure is now associated with a figure in the main text. This should help the reader access information in an easier, hierarchical manner.

**Reviewer #2 (Public Review):**
In this work, the authors investigated the pectoralis work loop and the function of the supracoracoideus muscle in the down stroke during slow flight in doves. The aim of this study was to determine how aerodynamic force is generated, using simultaneous high-speed measurements of the wings' kinematics, aerodynamics, and activation and strain of pectoralis muscles during slow flight. The measurements show a reduction in the angle of attack during mid-downstroke, which induces a peak power factor and facilitates the tensioning of the supracoracoideus tendon with pectoralis power, which then can be released in the up-stroke. By combining the data with a muscle mechanics model, the timely tuning of elastic storage in the supracoracoideus tendon was examined and showed an improvement of the pectoralis work loop shape factor. Finally, other bird species were integrated into the model for a comparative investigation.The major strength of the methods is the simultaneous application of four high-speed techniques - to quantify kinematics, aerodynamics and muscle activation and strain - as well as the implementation of the time-resolved data into a muscle mechanics model. With a thorough analysis which supports the conclusions convincingly, the authors achieved their goal of reaching an improved understanding of the interplay of the pectoralis and supracoracoideus muscles during slow flight and the resulting energetic benefits.

Thank you for your helpful and positive review. We appreciate that you summarized our manuscript accurately in a way that can help the reader.

**Recommendations For The Authors:**
The manuscript is very detailed and appears a bit long, including all the supplementary materials. It seems that the manuscript could easily have been separated into several publications, especially the comparative investigation including other extant bird species into the new model could have been a separate publication. This would have reduced the length of the supplements.

Thank you for your feedback on our manuscript; we made numerous improvements to improve the readability. Hence, we decided to not cut the supplement short or split it into more papers. We chose eLife because we wanted to publish this study in one complete manuscript. This has three benefits: (1) The reader can find all information in one well-edited paper at one publisher that is open-access and high-quality. (2) The first author works in industry and gets no benefits from publishing multiple papers, and hence he opted to publish one with support of the author team. (3) The senior author is not interested in fragmented publishing. Rather, he writes fewer, more comprehensive integrative papers because that is ultimately more informative for the reader: one trusted published source has all that is important to know based on this completed research project. Overall, we weren’t able to find technical information that shouldn't go in the paper using the lens of reproducibility, so the supplement is relatively long. Combining three methods (kinematics, forces, muscles), of which two are only available in the senior author’s lab, and extensive math (two new integrative models plus scaling laws) requires sharing the information needed for replication for all approaches we combine.

Also, some figure captions are very long and some of the content might have been included in the main text.

Complied: thank you for helping us streamline the captions. We reviewed all the figure captions and removed material that is repeated in the main text, but not essential to understanding the figures. However, because of the length of the manuscript and our desire to make the manuscript readable and clear, we left all other text in the captions intact so they remain readable independently of the main text. This way, the reader does not have to go searching for information in the main text just to make sense of the figures. This is especially important because readers often read the figures first before deciding if they want to read the main text completely. In addition, we moved two panels from Figure 2 into its associated figure supplement, because it was not a main point in the text, and hence this helped reduce the length of the caption in figure 2.